# Two-dimensional Inversion of wideband spectral data from the Capacitively Coupled Resistivity method - First Applications in periglacial environments

Jan Mudler[1], Andreas Hördt[1], Anita Przyklenk[1], Gianluca Fiandaca[2], Pradip Kumar Maurya[2], and Christian Hauck[3]

[1]Technische Universität Braunschweig, Institut für Geophysik und extraterrestrische Physik, Braunschweig, Germany
[2]Aarhus University, Department of Geoscience, Hydrogeophysics Group, Aarhus, Denmark
[3]University of Fribourg, Department of Geosciences, Fribourg, Switzerland

**Correspondence:** Jan Mudler (j.mudler@tu-bs.de)

**Abstract.** The DC resistivity method is a common tool in periglacial research, because it can delineate zones of large resistivities, which are often associated with frozen water. The interpretation can be ambiguous, however, because large resistivities may also have other causes, like solid dry rock. One possibility to reduce the ambiguity is to measure the frequency-dependent resistivity. At low frequencies (<100 Hz) the corresponding method is called induced polarization, which has also been used in periglacial environments. For the detection and possibly quantification of water ice, a higher frequency range, between 100 Hz and 100 kHz, may be particularly interesting, because in that range, the electrical properties of water ice exhibit a characteristic behavior. In addition, the large frequencies allow a capacitive coupling of the electrodes, which may have logistical advantages. The capacitively coupled resistivity (CCR) method tries to combine these logistical advantages with the potential scientific benefit of reduced ambiguity.

In this paper, we discuss CCR data obtained at two field sites with cryospheric influence: the Schilthorn massif in the Swiss Alps and the frozen lake Prestvannet in the northern part of Norway. One objective is to add examples to the literature where the method is assessed in different conditions. Our results agree reasonably well with known subsurface structure: At the Prestvannet site, the transition from a frozen lake to the land is clearly visible in the inversion results, whereas at the Schilthorn site, the boundary between a snow cover and the bedrock below can nicely be delineated. In both cases, the electrical parameters are consistent with those expected from literature.

The second objective is to discuss useful methodological advancements: First, we investigate the effect of capacitive sensor height above the surface and corroborate the assumption that it is negligible for highly resistive conditions. For the inversion of the data, we modified an existing 2-D inversion code originally developed for low-frequency Induced Polarization data by including a parametrization of electrical permittivity. The new inversion code allows to extract electrical parameters that may be directly compared with literature values, which was previously not possible.

# 1 Introduction

Electrical resistivity measurements determine electrical properties of the subsurface. They can support the investigation in periglacial environments, because they provide information on regions below the surface, otherwise only accessible by drilling. The DC resistivity method, called electrical resistivity tomography (ERT) if used to create vertical sections, is most useful to "detect, localize and characterise structures containing frozen material" (Hauck and Kneisel, 2008). The reason is that electrical resistivity dramatically increases when temperature falls below the freezing point of water. Therefore, ERT is "maybe the most universally applicable method in permafrost related mountain environments" (Hauck and Kneisel, 2008).

However, the interpretation of ERT data may be ambiguous, because the huge electrical resistivities associated with frozen material can also be due to dry, unfrozen rock, or to air in the pore spaces. In particular, when quantitative estimates, such as ice content, are desired, complementary information is usually required. One possibility is to combine ERT with other geophysical methods, such as ground penetrating radar, or seismics (Hauck et al., 2011). Another idea is to measure the frequency-dependence of electrical resistivity, along with the DC resistivity itself. In that case, the method is called induced polarisation (IP), or spectral induced polarisation (SIP), when measurements are made over a broad frequency range. The method has traditionally been used for a variety of applications, such as mineral exploration and the assessment of hydraulic properties of sediments, amongst others (Kemna et al., 2012). Applications in periglacial environments are sparse; a recent example of the investigation of a rock glacier is described in (Duvillard et al., 2018).

At the field scale, SIP measurements are typically made at relatively low frequencies (say <100 Hz). At higher frequencies (roughly > 100 Hz), field data are less frequently measured, one reason being that electromagnetic induction (EMI) may inhibit the determination of frequency-dependent electrical properties. In periglacial environments, where large resistivities are typically encountered, EMI is much less important, and the determination of electrical properties at the field scale might be feasible, as will be discussed further below.

Whereas at low frequencies, electrical properties are normally expressed by the imaginary conductivity (Kemna et al., 2012), electrical permittivity is often being used at higher frequencies, (e.g. Stillman et al., 2010). When using complex numbers, the use of conductivity or permittivity is mathematically are equivalent, but since the frequency dependence is caused by different physical processes in different frequency ranges, it is common to use permittivity for higher frequencies.

The frequency range >100 Hz up to several 100 kHz is particularly interesting for periglacial processes, because the permittivity of water ice exhibits a characteristic frequency-dependence in that range (Petrenko and Whitworth, 2002). A number of laboratory studies exists that investigate permittivity of natural material including ice (Olhoeft, 1977; Seshadri et al., 2008; Grimm et al., 2015; Murton et al., 2016; Zorin and Ageev, 2017), suggesting the idea that ice content might even be determined quantitatively (Bittelli et al., 2004). Therefore, if permittivity could be measured at the field scale, a unique piece of information would be contributed that can help to reduce the ambiguity that exists when only DC resistivity is measured.

The usage of relatively high frequencies can help to overcome another major problem associated with electrical measurements in periglacial environments: the coupling between the electrodes and the often hard and very resistive surface (Hauck and Kneisel, 2008). At large frequencies, capacitive coupling becomes feasible. Instead of skewers, plate electrodes may be used.

They form a capacitor with the ground, and allow contact-free injection of current even for extremely resistive surfaces (Hördt et al., 2013). In that case, the method may be considered an extension of high-frequency SIP, called capacitively coupled resistivity (CCR).

The CCR method has originally been suggested for applications on space missions (Grard, 1990; Grard and Tabbagh, 1991), where the conditions (large resistivities, difficult electrode coupling) may be similar to those in periglacial environments. Besides application in space (Seidensticker et al., 2007), devices have been developed for investigations in urban areas such as facades (Souffaché et al., 2010) or roads (Dashevsky et al., 2005; Flageul et al., 2013), as well as archeological sites (Tabbagh et al., 1993) and environmental problems (Kuras et al., 2007).

The first high-frequency SIP measurements at the field scale in periglacial environments were carried out by Grimm and Stillman (2015), who used the method for a characterisation of subsurface ice. Przyklenk et al. (2016) discuss data from CCR measurements using data acquired on an ice layer at Zugspitze mountain, and develop an inversion scheme based on a homogeneous halfspace assumption.

Although the concept of obtaining high-frequency SIP data using CCR may be considered proven, there is still little experience with field data, and several open questions on the applicability remain. One aspect is the sensor height effect: a distortion of the data arising from electrodes being in a finite distance from the ground. The effect has been investigated both theoretically and experimentally (Kuras et al., 2006; Wang et al., 2016), and there are indications that it can be neglected in periglacial applications characterized by large resistivities (Przyklenk et al., 2016). However, the effect depends on the specific conditions in each survey area, and there is little practical experience.

A second aspect not fully solved is the inversion. Przyklenk et al. (2016) used a so-called single site inversion that treats each 4-point measurement individually assuming a homogeneous halfspace, and inverts only the spectral behaviour. This was justified by the homogeneous subsurface and the small spatial coverage of that data set. Grimm and Stillman (2015) investigated several methods of 2-D inversion in order to produce a vertical cross section. The challenge was that existing IP inversion codes, such as developed by Routh et al. (1998) and Kemna et al. (2000) were only able to invert single frequency data, and postprocessing is required to integrate all spectral and spatial data. Grimm and Stillman (2015) discuss some difficulties they encountered, which they finally circumvented using the time-lapse feature of RES2DINV, a widely used 2-D inversion code for DC resistivity and IP data (Loke and Barker, 1996). Recently, 2-D inversion codes have become available which are able to invert all frequencies and spatial data points at the same time (Günther and Martin, 2016; Maurya et al., 2018).

Here, we discuss two case histories of CCR applications in periglacial environments, one from the Schilthorn massif in the Swiss Alps and the other one from the frozen lake Prestvannet in the northern part of Norway. Besides the general usefulness of gaining experience with CCR field applications and extending the sparse data set existing in the literature, we focus on two aspects. For the 2-D inversion, we modified the SIP inversion code AarhusInv (Auken et al., 2014), an inversion tool for various geophysical methods, to consider the frequency-dependence and apply the code to the data of our two test sites. We also investigate the potential effect of electrode height and show that it is negligible in both cases.

The results of the 2-D inversion will be compared with existing knowledge about the subsurface stratifications and materials. Although a quantitative assessment of the parameters is difficult because of the sparse availability of additional information, we

show that the results are at least not implausible. The new inversion code is suitable for field data and constitutes one step forward towards the ultimate goal: reducing the ambiguity in the interpretation of resistivity data and maybe provide quantitative information, such as ice content.

## 2 Measurements and test sites

For the application of CCR, we focus on the cryosphere (i.e. ice, snow, permafrost), where the logistic advantages of the capacitive coupling are given in terms of highly resistive ground and in some cases hard surfaces (e.g. ice or frozen ground). The method enables the ability to measure directly on snow and ice. The measurements were carried out using the Chameleon equipment from Radic Research, which is specifically designed for the application of broadband measurements of the electrical impedance (Radić, 2013; Przyklenk et al., 2016). The prototype device uses a 4-electrode array. Therefore, two-dimensional

measurements along a profile and in depth are achieved by gradually shifting and enlarging the array. It is possible to measure in a range from $1\,\mathrm{Hz}$ up to $240\,\mathrm{kHz}$ at 19 discrete frequencies. The results are the spectral values of the magnitude $|Z(f)|$ and the phase shift $\varphi(f)$ of the impedance. Wenner- and dipole-dipole configurations were used.

### 2.1 Schilthorn

The survey was carried out in July 2016 on the Schilthorn massif, in the Bernese Alps, Switzerland. There is occurrence of

alpine permafrost in the area (e.g. Hilbich et al., 2008; Scherler et al., 2010). Figure 1 illustrates the geographical location. Panel B shows the area from village Mürren up to the summit Schilthorn. The mountain station Birg is in between and can be reached by a cable car. The position of the selected profile B-SCH, north of Birg, at an altitude of about $2700\,\mathrm{m}$ a.s.l. and with a length of 27 meters, is shown in Fig. 1C. The surface in this area mostly consists of rock, which is covered with snow most parts of the year (i.e. October-July). On the summit area, the ground material is described as weathered. The occurence of an

ice layer under the snow is possible, as modelled by Scherler et al. (2010).

The photograph in Fig. 2 shows an example of the equipment layout in the field. The plate electrodes, which are covered with capton foil for galvanic decoupling, were arranged in a profile line. They are connected by cables through a probe and a remote unit to the base unit (Przyklenk et al., 2016), which controls the measurements. The surface at the time of the measurements was covered by a layer of snow, which was frozen on the top. The depth of this snow layer was separately measured every

two meters using a dipstick for later validation of the results. Measurements were done along the profile in a dipole-dipole configuration ($a = 1\,\mathrm{m}$) for several electrode spacings ($n = 1 - 6$). They were not carried out with the same electrode spacing throughout the profile. Measurements with wider electrode spacing and corresponding larger penetration depth were carried out only on the first half of the profile.

### 2.2 Tromsø

The measurements in Norway were done in 2015 on the frozen lake Prestvannet near the town of Tromsø. Figure 3 shows the geographical position of the area and the test site. In Fig. 3B, a part of the peninsula Tromsøya, with the city Tromsø and the

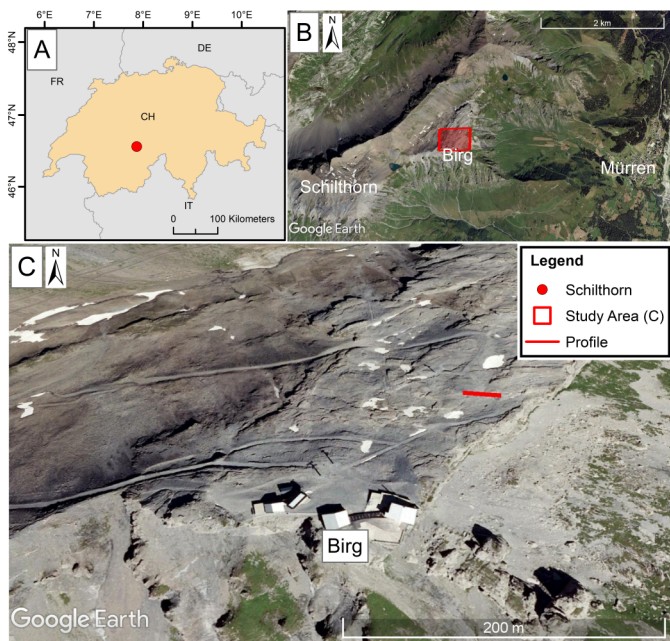

**Figure 1.** Geographical maps of the Schilthorn area. Panel A shows the location in Switzerland. In panel B the Schilthorn area with the village Mürren, the mountain station Birg and the summit is shown. The area around Birg, where the measurements took place is shown in panel C, including location of profile B-SCH.

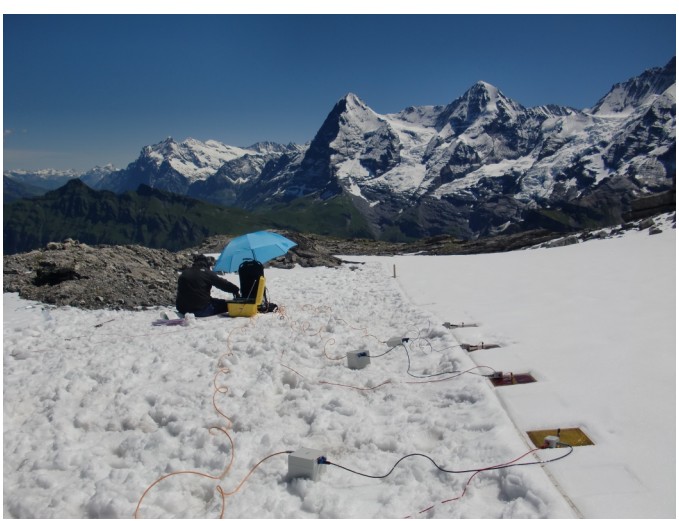

**Figure 2.** Photograph of the measurements at profile B-SCH (Schilthorn area) in July 2016 with the Chameleon measurement device. The four plate electrodes are lying in line on the snow surface. The larger yellow box is the base unit which is connected by cables to the electrodes with the cubic grey remote units in between.

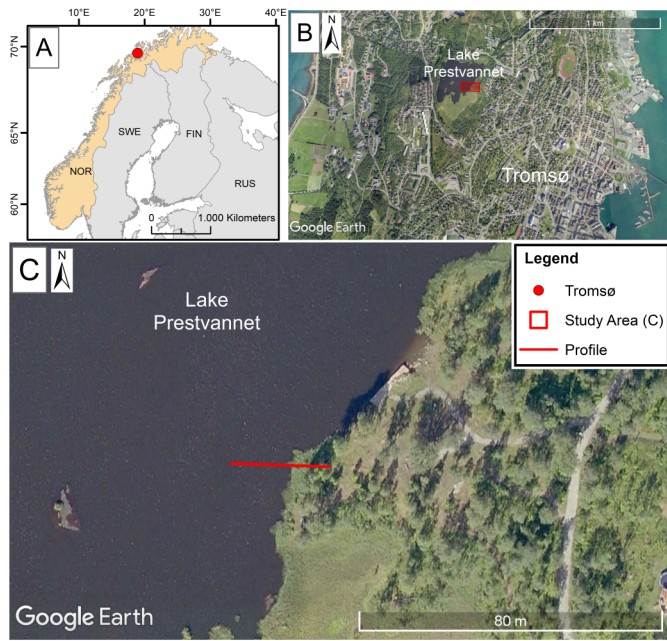

**Figure 3.** Geographical Maps of Lake Prestvannet. The location in the northern part of Norway is shown in part A by the red dot. The lake is located on the peninsula Tromsøya, presented in panel B, close to the city Tromsø. A detailed view of the test site is given in panel C, showing the profile crossing the boundary from the lake to the shore.

lake are visible. Lake Prestvannet covers an area of about 10 ha, has a maximum depth of 4 meters and is covered by ice most of the year (Stabbel, 1985). Although no quantitative statement is made, investigations of the lake water qualitatively indicate a high salinity (https://memim.com/prestvannet.html). The part of the lake, where the test site is located, is shown in panel C, including the profile, which has an extension of 33 meters length and crosses the shore of the lake. The shore was covered with a layer of snow, the lake itself was frozen, and measurements took place directly on the lake ice. Starting the profile on the lake and ending at the shore, the transition to the lake surface is at about profile coordinate $20.5\,\mathrm{m}$.

The measurements were done with a fixed electrode spacing in Wenner configuration ($a = 1.5\,\mathrm{m}$) to investigate differences in the measured data due to the sub-vertical lake-shore-boundary. The penetration depth of the measurements has therefore a maximum of $1.5\,\mathrm{m}$ (Militzer and Weber, 1985). Additional measurements indicated that the boundary to the liquid water was at a minimum of $4\,\mathrm{m}$ depth, below the penetration depth of the data.

## 3    Basics of the Capacitively Coupled Resistivity Method

When a time-varying current is injected into the ground, two different physical mechanisms are stimulated: the conduction current associated with the electrical resistivity, and the displacement current controlled by the electrical permittivity. The response of the material contains a combination of these two mechanisms, which can be measured as the impedance. Consequently, it is

possible to define a complex value named the effective conductivity, respectively the effective permittivity $\varepsilon^*$, which combines the conduction and polarization properties, as:

$$\varepsilon^*(\rho, \varepsilon_r, \omega) = \varepsilon_0 \varepsilon_r + \frac{1}{i\omega\rho}, \tag{1}$$

where i is the imaginary unit, $\varepsilon_0$ the permittivity of the vacuum and $\omega$ the angular frequency. In the most general form, both the electrical resistivity $\rho$ and the relative electrical permittivity $\varepsilon_r$ may be considered complex and frequency-dependent (Loewer et al., 2017). However, since this general description is redundant, it is a matter of choice or convention whether frequency-dependence is expressed by resistivity, respectively conductivity, or permittivity. In the low-frequency range used by conventional SIP, frequency-dependence is commonly expressed by an imaginary conductivity (e.g. Maurya et al., 2018), whereas at higher frequencies permittivity is normally used (e.g. Stillman et al., 2010). Here, we choose to express our results in terms of frequency-dependent permittivity, assuming resistivity a real, constant value, as will be detailed below.

The three variable quantities, i.e. $\omega$, $\rho$ and $\varepsilon_r$ determine in mutual dependence the weighting of the two current components. The injected current and the measured voltage are in-phase for the proportion of conduction current and shifted by $-90°$ for the displacement current component.

Most geophysical methods working with time-varying electric fields focus on one of the two mechanisms by defining a chosen frequency range and neglect the other part. The ratio of the proportions of both current mechanisms gives an estimation for the physical regime of the measurements. Geophysical methods such as Induced Polarization (IP) or Magnetotellurics (MT) (Telford et al., 1990) work in a rather low frequency range where the conduction current dominates the signal of the impedance. The Ground Penetrating Radar (GPR), on the other hand, works at very high frequencies and focuses on the displacement current, determining the permittivity (Zorin and Ageev, 2017). Our aim is to measure in an intermediate frequency range where both current mechanisms are relevant in order to determine both electrical parameters (see Eq. (1)). In order to ensure this in our given frequency range, the subsurface materials have to exhibit relatively high resistivities and permittivities. The condition may be calculated based on the loss tangent (Przyklenk et al., 2016). Typically, for our frequency range the resistivity has to be greater than $1000\,\Omega$m. The needed conditions are particularly prevalent in periglacial areas, with the occurrence of ground ice (Arenson et al., 2015; Hauck and Kneisel, 2006).

The description of the complex impedance by Kuras et al. (2006), modified by Przyklenk et al. (2016) in order to use the unmodified geometry factor $K$, known from DC resistivity, is:

$$Z(\omega, \rho, \varepsilon_r, h) = \frac{1}{2i\omega\varepsilon_0 K}[1 - \alpha(\rho, \varepsilon_r)H(h)], \tag{2}$$

where the reflection factor $\alpha$ contains both electrical parameters of the subsurface.

Special attention is given to the height factor $H(h)$. It depends on the geometry factor $K$ and a vertical geometry factor, which describes the height of the capacitively coupled sensors. In the case of an ideal contact of the electrodes to the ground, the

height $h$ becomes zero and the resulting height factor $H$ becomes one. Thus, the electrical parameters for each frequency can be determined directly from the real and imaginary part of the impedance (Przyklenk et al., 2016):

$$\rho = \frac{K}{Re(Z^{-1})} \tag{3}$$

$$\varepsilon_r = \frac{Im(Z^{-1})}{K\omega\varepsilon_0} - 1 \tag{4}$$

The challenge is that the electrodes, especially in the case of plates or discs usually do not rest over their entire surface on the ground. Rather, with a slight unevenness of the ground, a contact of the electrodes to the ground is ensured only at a few points.

This results in a mean non-zero height $h$ of the electrode surface over the ground, which however can hardly be measured directly. The height dependence was already discussed by a few authors (Kuras et al., 2006; Przyklenk et al., 2016). Even small heights in the range of micrometers can cause large differences in the measured impedance, but this dependence becomes weaker as resistivity and permittivity increase.

### 3.1 Cole-Cole Model

The electrical permittivity and the resistivity are not constant values in most cases but vary with frequency. Polarizable materials, e.g. water-saturated sediments or mineralized rocks exhibit a strong frequency dependence of electrical parameters (Zorin and Ageev, 2017). This is especially true in periglacial areas, for materials with pure ice or large ice contents (Petrenko and Whitworth, 2002; Bittelli et al., 2004; Stillman et al., 2010). Przyklenk et al. (2016) investigated several parametrizations of the frequency-dependence of resistivity and permittivity. They suggest the use of the Cole-Cole Model (CCM) (Cole and Cole,

1941), which provides reasonable results when fitting the spectral data of CCR measurements. For variable data with more spectral shape, a dual CCM, corresponding to a model of a two-component mixture, might be necessary for the evaluation of the impedance spectrum. For our studies, we decided to use the single CCM, which includes just one material, because it can fit our data with a minimum number of parameters. For the relative complex permittivity, the single Cole-Cole Model is expressed by:

$$\varepsilon_r^* = \varepsilon_{HF} + \frac{\varepsilon_{DC} - \varepsilon_{HF}}{1 + (i\omega\tau)^c} + \frac{1}{i\omega\varepsilon_0\rho_{DC}}. \tag{5}$$

The description of the frequency dependence of the electrical parameters is based on five Cole-Cole parameters: the DC resistivity $\rho_{DC}$, a low-frequency limit $\varepsilon_{DC}$, a high-frequency limit $\varepsilon_{HF}$, which is referred in the literature as the dielectric

constant, the relaxation time $\tau$ and the relaxation exponent $c$. The positive relaxation exponent can range up to a maximum value of one, for which the model simplifies to the Debye Model (Petrenko and Whitworth, 2002). The parameters are directly

related to a physical context. Thus, they are material-specific parameters for which ranges of literature values are known, that can be used for a discussion of the inversion results.

## 3.2 Operating range

There is a parameter range in which CCR method is feasible in the sense that the underlying assumptions are fulfilled and the physical process that is used to determine the spectral behaviour of permittivity and conductivity actually dominates. The term "parameter range" refers to the frequency, the spatial scale (i.e. distance between transmitter and receiver), and electrical conductivity and permittivity. The CCR method operates in the „geometric sounding" range, where the investigated volume and in particular the penetration depth depend only on the location and the distance between transmitter and receiver. This is the same condition that applies to the ERT method.

The two processes that need to be investigated because they may limit the operating range of CCR are electromagnetic induction (EMI) and wave propagation. Electromagnetic induction currents are caused by the time-varying magnetic fields which always co-exist with electric fields. Several methods are based on EMI, which is particularly important if the conductivity is large. Wave propagation is the basis of ground penetrating radar (GPR) methods, and is particularly important for very large frequencies. For an assessment of the relative importance of the processes we use the consideration by Weidelt (1997)), who compares the wave lengths (or their inverse, the wavenumbers) of the three processes with each other. The wave length of EMI is equal to the skin depth, given by

$$\delta = \sqrt{\frac{2\rho}{\omega\mu}} \tag{6}$$

where $\mu$ is the magnetic permeability, where in this context it is sufficient to use the vacuum value, which is $\mu_0 = 4\pi \times 10^{-7} \frac{Vs}{Am}$. The wave length of wave propagation is given by

$$\lambda = \frac{2\pi}{\omega\sqrt{\varepsilon\mu}}. \tag{7}$$

For geometric sounding, the signal is composed of many different wavelengths, but for an estimate of the relative importance of the processes, it is feasible to use an appropriate measure of the spatial scale of the electrode configuration (i.e. the electrode distance a for Wenner configuration), as the dominant wave length. The equation that allows to compare the three processes is (Weidelt, 1997):

$$\gamma^2 = \frac{4\pi^2}{a^2} + \frac{2i}{\delta^2} - \frac{4\pi^2}{\lambda^2} = G + EMI - WP \tag{8}$$

where gamma is the complex wavenumber including all three processes, and the symbols G, EMI and WP stand for the

geometrical sounding, EM induction and wave propagation term, respectively. The process corresponding to the largest term will dominate, and the other terms may be negligible, depending on their magnitude.

Since there are four parameters controlling Eq.(6)-(8), which also depend on each other (i.e. permittivity depends on frequency), it is difficult to give an operating range of general validity. It seems more feasible to evaluate the terms in Eq.(8) for specific conditions. For the measurements discussed in this paper, we can carry out the following estimates: The maximum frequency of our system is $240\,\mathrm{kHz}$. Assuming a minimum relative permittivity of 3, the minimum wavelength of wave propagation given by Eq.(7) is $722\,\mathrm{m}$. The minimum wavelength of EMI given by Eq. (6) depends on electrical resistivity. Assuming $\rho = 100\,\Omega\mathrm{m}$ as the minimum resistivity, which is a little smaller than the minimum values actually encountered, we obtain approx. $10\,\mathrm{m}$ for the minimum EMI wavelength. From Eq.(8), it is clear that the geometrical sounding term will be minimum for the largest electrode spacing, and therefore we use our largest spacing $a = 1.5\,\mathrm{m}$ in Eq.(8) to obtain a worst-case estimate.

With these numbers for the wave lengths, we obtain for the magnitudes: $G = 17.5$; $EMI = 0.02$; $WP = 7.6 \cdot 10^{-5}$. Therefore, the wave propagation term is by far the smallest and can safely be neglected, the EMI term is still significantly smaller than the geometrical sounding term. We therefore assume that for the data discussed here, EMI is also negligible, in case of high resistive environments and small electrode distances (Fiandaca, 2018). Nevertheless, we are also aware that a comprehensive assessment requires precise modelling of the equations, which will be subject of future work. If we chose a larger electrode spacing, $a = 30\,\mathrm{m}$, then $EMI = 8$, this would be in the same order of magnitude as G, and we would have to carry out a more thorough analysis.

In addition to these purely physical considerations, we also have to take technical issues into account, such as coupling of the capacitive electrodes, and electromagnetic coupling between cables and the ground, which are less simple to estimate or to correct for. Therefore, we consider it necessary to gain experience and to test the entire system under a variety of conditions.

### 3.3 Single Site Inversion

The single site inversion was the primary method used in Przyklenk et al. (2016). In this spectral approach, the data of each measured 4-point array is inverted separately, i.e. without influence of other measurements. Under the conditions of geometrical sounding, the penetration depth of the measurement is only controlled by the geometric size of the configuration, i.e. the geometry factor $K$, in contrast to the frequency sounding where the frequency dependent skin depth describes the penetration depth (McNeill, 1980).

The fit of the measured spectral data of the magnitude $|Z(f)|$ and phase shift $\varphi(f)$ is done under parametrisation of the complex impedance (Eq. (2)) by the single Cole-Cole Equation (Eq. (5)). The inversion is based on the model of a homogeneous halfspace. From the result of the inversion, the five Cole-Cole Model parameters can be extracted. Moreover, the single site inversion has the possibility to take the sensor height effects into account. By using the mean electrode height $h$ as an additional free inversion parameter, it is possible to include the effect of electrode height and at the same time determine its value. Thereby, capacitively coupled measurements taken under conditions of strong height influence, in particular on low resistive subsurfaces, can also be evaluated. If the height is neglected during inversion, this can in principle lead to a distortion of the data and erroneous results. Since in the next step we use a conventional 2-D inversion code for spectral IP data, where the height

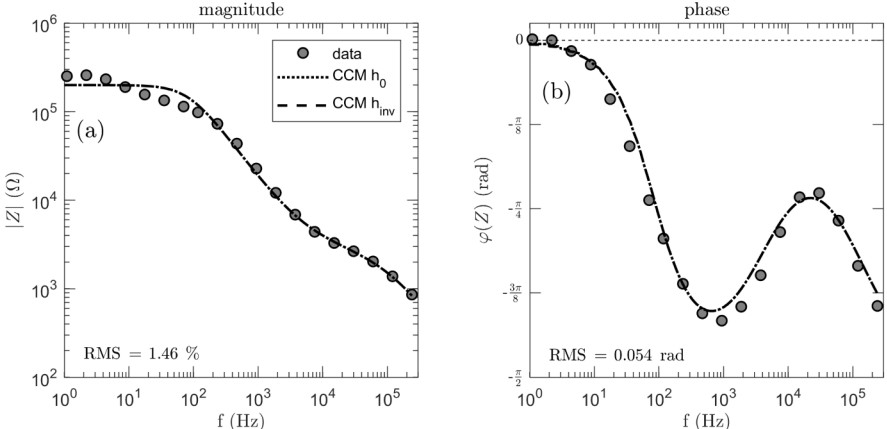

**Figure 4.** Spectra for the magnitude (a, left) and the phase shift (b, right) of the impedance for measured data (dots) and inversion results by using the Cole-Cole Model (CCM, lines). The inversion was carried out twice, with the assumption of zero electrode height ($h_0$) and by calculating the height as an additional inversion parameter ($h_{inv}$). The simulated curves of the two inversions can not be distinguished from each other. Data were measured at profile B-SCH, Schilthorn area, Switzerland, in a dipole-dipole configuration ($a = 1m$, $n = 1$). The CCM parameters are ($h_0$ and $h_{inv}$): $\rho_{DC} = 3.8 \cdot 10^6 \,\Omega m$, $\varepsilon_{DC} = 53$, $\varepsilon_{HF} = 2.8$, $\tau = 3.6 \cdot 10^{-5}\,\mathrm{s}$, $c = 0.82$.

effect is not included into the forward modelling, it is essential to test whether it is justified to neglect it. For this purpose, we carry out the single site inversion twice: first including the height effect by setting the height as a free parameter and once under the assumption of no sensor height, i.e. by fixing it to zero.

Figure 4 shows a representative example of the spectra for magnitude (a) and phase shift (b) from a measurement at profile

B-SCH. The points indicate the measured data, the lines are the calculated spectra for the best-fit model. Inversion is done with (CCM $h_{inv}$) and without (CCM $h_0$) determination of height. The calculated height for CCM $h_{inv}$ is $7 \cdot 10^{-7}$ m. This is so close to zero that the results exhibit no visible difference in the measured frequency range. The parameters of the Cole-Cole Model, given in the caption of the Figure, have no difference if rounded to a maximum of two decimal places.

The magnitude adopts a value of around $2 \cdot 10^5\,\Omega$ for low frequencies, where the curves converge to a constant value. The

phase shift covers almost the entire range that is theoretically possible (0 to $-90°$). The values close to zero for low frequencies indicate a domination of conduction currents, whereas the deviation from zero for increasing frequencies indicates the increasing relevance of displacement currents. It is expected that for higher frequencies out of the measured range, the phase shift approaches the limit of $-90°$ where displacement currents dominate. The aim of measuring the intermediate range with the transition between both current mechanisms has been achieved in this case. The measured spectral signals in Fig. 4 are

typical of the whole measurements on the profile. The values of magnitude are decreasing for larger configurations because of the increasing geometry factor. The phase shift shows the characteristic wavelike shape with a local minimum and maximum.

The data fit of the single site inversion is reasonable, justifying the usage of the single Cole-Cole Model.

In case of Lake Prestvannet we assume a sub-vertical separation through the measurements over the lake shore. Two representative signals are shown in Fig. 5 to illustrate the difference between the characteristic curve shapes. The magnitude exhibits a stronger frequency dependence in case of the land measurements and a higher value for low frequencies of about one order compared to the lake measurements. For the phase shift, the shape of the curve measured on the lake is flatter and the local minimum is at higher frequencies. The phase shift shows smaller dynamics for the lake than for the land measurements. The Figure illustrates that the different ground materials provide significant differences in their response.

The inversion was done with and without including the determination of sensor height. For the onshore measurement the estimated height is $2.5 \cdot 10^{-5}$ m, which results in no visible difference between the simulated curves. For the lake measurement the fitted height of $1.2 \cdot 10^{-3}$ m leads to a visible difference for the phase shift (panel b). The two calculated spectra vary for the lowest frequencies. While the inversion with zero height (continuous line) converges towards zero, the version including height (dashed line) shows a deviation from zero consistent with the data. The data fit shows a difference also in terms of the root mean square (RMS) which is better for CCM $h_{inv}$. It is known from theory that the height dependence has stronger effects for lower frequencies and is stronger for the phase than for the magnitude of the impedance (Przyklenk et al., 2016). Our data, with a visible effect in the phase shift and negligible effect in the magnitude for the data set with the smaller impedance (panel a,b) is consistent with these theoretical results.

## 3.4 Influence of Electrode Height on Cole-Cole Parameters

In the following, we analyse the dependence of the Cole-Cole parameters on the electrode height in some more detail. The reason is, that the 2-D inversion used later is not able to consider non-zero electrode height. Therefore, the single site inversion was performed for every measured array of both field areas in both versions, with and without determining $h$. For every array, the five resulting Cole-Cole parameters were put into relation for both inversions and the mean deviation in percent was calculated. This deviation is shown in dependence on the estimated sensor height in Fig. 6.

For larger sensor height, the Cole-Cole results are more affected. The Schilthorn data (red dots) show a strong increase of deviations from approximately $5 \cdot 10^{-3}$ m height on. For the Tromsø data, which are shown seperately for the measurements on the lake (blue dots) and on shore (yellow dots), this increase occurs for sensor heights about one order of magnitude lower. The different behaviour can be explained by the condition of lower electrical resistivities for the Tromsø measurements. As mentioned earlier, the height effect is stronger for lower ground parameters of resistivity and permittivity. Furthermore, for the Schilthorn data higher values of sensor height were determined. This could indicate that in case of the solid ice surface, as on the lake, the contact of electrodes to the ground is very smooth, resulting in a more homogeneous sensor height. On the other hand, the snow surface at Schilthorn builds a more porous ground. The loose material might cause a poorly defined contact and lead to an artificially increased apparent electrode height. The Schilthorn data shows scattered sensor heights, which indicates the uncertainty in the electrode contact surface. It should be noted that the calculated sensor height in Fig. 6 are shown only down to the lowest values of $5 \cdot 10^{-6}$ m, but also lower values were determined. Investigations from Przyklenk et al. (2016) and Kuras et al. (2006) indicate that electrode heights lower than around $10^{-4}$ m do not effect the measured signal, especially

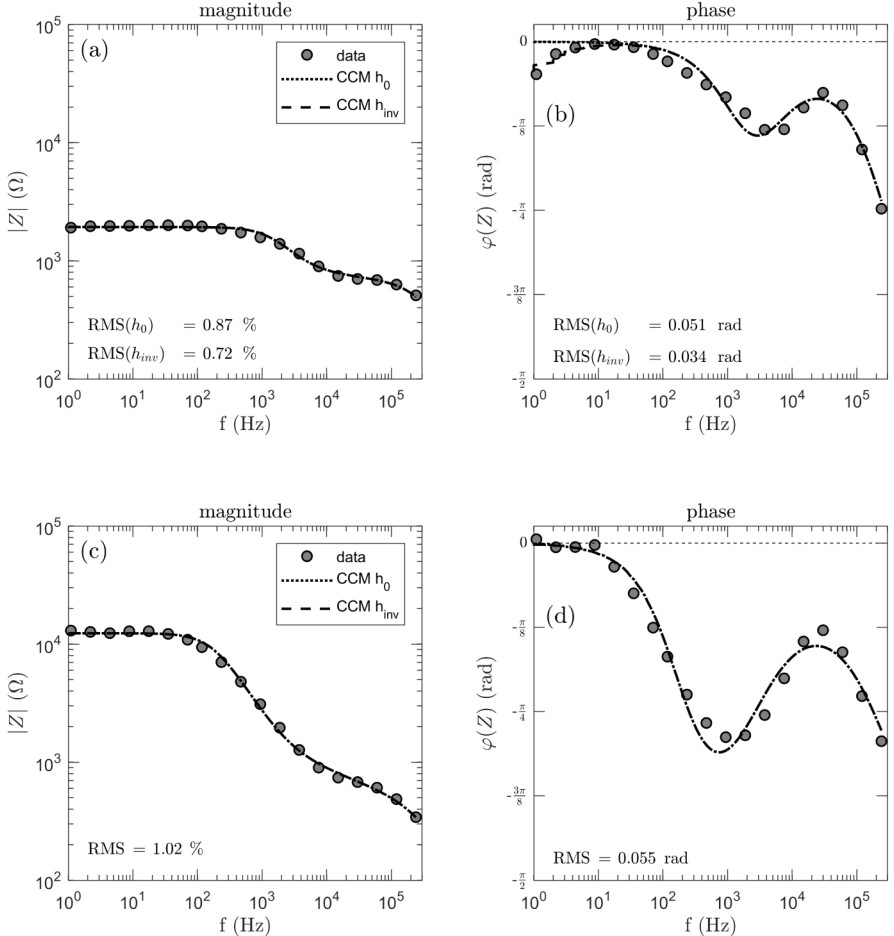

**Figure 5.** Spectra of two measurements from the Tromsø site. Panel (a,b) show data from the lake and panel (c,d) data from a measurement on shore. The figure shows the measured data and the inversion results using the Cole-Cole Model. The simulated curves with zero electrode height ($h_0$) and inverted height ($h_{inv}$) can not be distinguished, except for the phase shift of the lake measurement (panel b). Both measurements were taken with a Wenner-configuration with $a = 1.5\,\mathrm{m}$.

The CCM parameters for the lake measurement are ($h_0/h_{inv}$): $\rho_{DC} = 1.82 \cdot 10^4/1.81 \cdot 10^4\,\Omega\mathrm{m}$, $\varepsilon_{DC} = 374/370$, $\varepsilon_{HF} = 8.80/8.83$, $\tau = 4.2 \cdot 10^{-5}/4.1 \cdot 10^{-5}\,\mathrm{s}$, $c = 0.93/0.93$.

The CCM parameters for the onshore measurement are ($h_0$ and $h_{inv}$): $\rho_{DC} = 1.2 \cdot 10^5\,\Omega\mathrm{m}$, $\varepsilon_{DC} = 670$, $\varepsilon_{HF} = 12.8$, $\tau = 7.1 \cdot 10^{-5}\,\mathrm{s}$, $c = 0.84$.

under high resistive conditions. Smaller heights determined by the inversion are mainly caused by numerical reasons and does not represent physical conditions. They can be seen as equal to zero. The determined values of the Schilthorn measurements

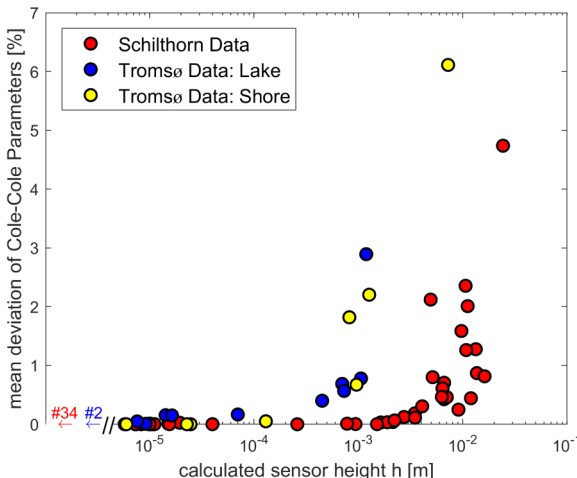

**Figure 6.** Mean deviation of Cole-Cole parameters vs. sensor height. The deviation was calculated by performing the single site inversion with and without the determination of sensor height and calculating the ratio for all the five Cole-Cole parameters. The data are separately shown for all measurements at profile B-SCH (Schilthorn, red) and for Lake Prestvannet (Tromsø) on the lake (blue) and on shore (yellow). Values for sensor heights lower than $5 \cdot 10^{-6}$ m are not displayed but their number of measurements is shown.

range from less than $10^{-9}$ m up to centimeters. On the other hand the Tromsø results only vary from $10^{-7}$ m to the millimeter range.

Additional investigations, which will not be further elaborated here, have indicated that the parameters $\rho$, $\varepsilon_{DC}$ and $\tau$ are generally more affected than $\varepsilon_{HF}$ and $c$. That is consistent to the fact that the electrode height effects mainly the lower

frequencies. All in all, the deviations of the Cole-Cole results are relatively small. Except for the two highest values in Fig. 6, their effect is smaller 3 percent. These deviations are considered to be acceptable, compared to the typical Cole-Cole parameter resolution for inversions (Yuval and Oldenburg, 1997; Madsen et al., 2017). The results justify therefore the use of inversions without considering effective sensor height, as in the case of the 2-D inversion (see next section). It should be noted that under less favourable conditions, depending on the electrical parameters of the soil and the texture of the surface, neglecting

the height can lead to larger errors. Neglecting height is therefore not a general recommendation, but has to be investigated separately for each application with different subsurface conditions. In the case of snow or icy ground, one additional benefit is that these are rather smooth surfaces, where the electrode height is small. On uneven surfaces, such as gravel and rock fields, an installation of the plate electrodes without height variations may be difficult to achieve.

## 4    2-D Inversion with AarhusInv

The full spectral inversion of complex resistivity data, where all frequencies are being inverted simultaneously, has been a challenge for some time. For example, Grimm and Stillman (2015) used a workaround based on the time-lapse feature of

RES2DINV (Loke and Barker, 1996) to invert their broadband SIP data from a periglacial environment. Recently, a few codes for full spectral inversion have become available (Günther and Martin, 2016; Maurya et al., 2018). Here, we use the program AarhusInv (as in Maurya et al., 2018), which is a tool for the inversion and modelling of geophysical data for several measurement methods (Auken et al., 2014). The software is freely available for non-commercial purposes. In AarhusInv the complex impedance is modelled in 2-D solving the Poisson's equation, Fourier transformed in the strike direction, without considering EM effects (Fiandaca et al., 2013). All the inversion parameters are inverted simultaneously using all the measured frequencies in a unique inversion process (equivalently to the spectral full-decay inversion of the time-domain IP data). In case of Induced Polarization, where the frequency range is usually only up to one kilohertz, in general relatively small phase shifts are measured. In order to use the inversion for the CCR method, we included the permittivity Cole-Cole Model defined by Eq. (5) to parameterize the frequency-dependent electrical properties. Compared to the conventional Cole-Cole resistivity model that is defined by four parameters and is sometimes used to parameterize low-frequency SIP spectra (Pelton et al., 1978; Tarasov and Titov, 2013), the model defined by Eq. (5) has one more parameter, basically corresponding to the high-frequency limit of permittivity. Therefore, the result of the inversion is a distribution of the five Cole-Cole parameters. As discussed previously, the height of the electrodes is not included into AarhusInv and assumed to be zero. Through this approximation the application for CCR data is usually just suitable under highly resistive conditions. Of course, this inversion method could generally be used for high frequency spectral resistivity measurements, including galvanically coupled electrodes.

In the following, we will show the results of the 2-D inversion for both field sites. The resulting distribution of all Cole-Cole parameters will be discussed and compared with the expected properties of the subsurface.

## 4.1 Schilthorn

The measured data from profile B-SCH were evaluated by the 2-D inversion. The result is shown in Fig. 7, where several 2-D models for the five Cole-Cole parameters $\rho$, $\varepsilon_{DC}$, $\varepsilon_{HF}$, $\tau$ and $c$ are shown color-coded vs. depth and horizontal coordinate. The dashed black line corresponds to the manually measured depth of the top snow layer.

In this context, it is important to discuss the resolution of the Cole-Cole parameters, because it is known from previous studies that it can be difficult to reliably estimate relaxation time and frequency exponent (e.g. Madsen et al., 2017). The problems usually arise if the frequency where the phase peak occurs, which is related to the relaxation time, is outside the measured acquisition range. Weigand and Kemna (2016) discuss similar observations when average parameters are derived from spectral decomposition techniques. The particular benefit of our acquisition system is the wide frequency range compared to conventional SIP system. As a result, all phase peaks corresponding to the relaxation times are in fact being measured. Therefore, we are confident that the Cole-Cole parameters of the inversion results are well determined. Nevertheless, we carried out additional experiments where we fixed the frequency exponent c, as suggested by Weigand and Kemna (2016). In that case, the data fit deteriorates, in the sense that the curve shape vs. frequency cannot be matched that well any more. We consider this as evidence that even c is not poorly constrained. The strong variability of c observed in the inversion results may be justified by the actual change in materials. Finally, we rely on the depth of investigation (DOI) as an objective measure in which regions parameters are well constrained. The calculation of the DOI was described in Fiandaca et al. (2015). The brighter areas in Fig. 7 are those

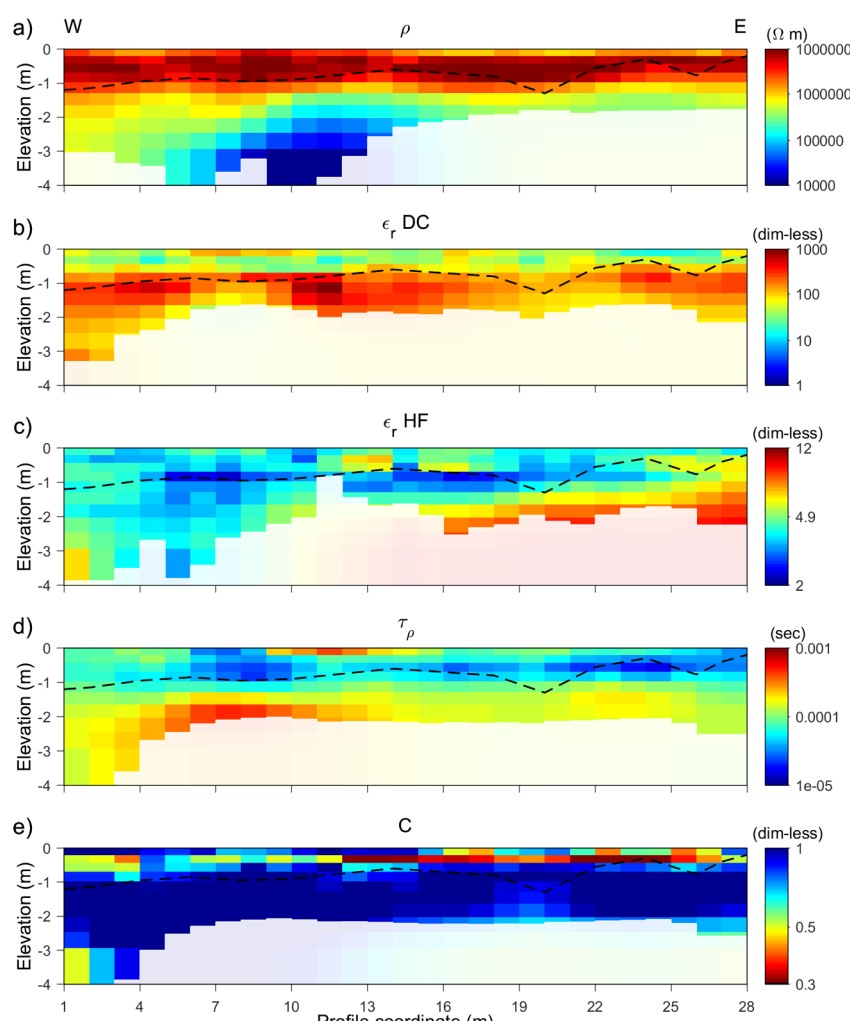

**Figure 7.** Result of the AarhusInv 2-D inversion for the data from the Schilthorn area along the profile B-SCH denoted in Fig. 1. The figure shows the sections of the five Cole-Cole parameters (a-e) defined by eq. (5). The dashed line shows the separately measured depth of the snow-layer, the brighter parts represents the area where the depth of investigation is exceeded.

below the DOI, areas where the parameters can no longer be reliably estimated. This boundary differs for each parameter, having in most cases the deepest extent for the resistivity model. We are aware that other tools to assess parameter resolution may exist, but a comprehensive treatment of this subject is beyond the scope of this paper.

First, we focus on the structural aspects of the results. The structure is a little different for each of the five sections, but in general a structure of two layers can be recognized, representing the top snow layer and the underlying surface layer. Because of the additional snow depth measurements, the results of the inversion can be validated. Especially for $\varepsilon_{DC}$, the boundary of the snow layer agrees well with the corresponding parameter contrasts. Around profile meter 20, where the boundary indicates

a ditch, it becomes particularly clear how precisely the layer structure is reflected by the low-frequency permittivity value (panel b). The layer boundary can also be seen from the result of the resistivity (panel a), where a highly resistive surface layer is followed by a more conductive material. Unlike in the $\varepsilon_{DC}$ section, there is a continuous decrease in the value with the depth, which makes the layer transition appear more smoothly. The layer boundary is also apparent in the section of parameter

$\tau$ (panel d). The relaxation exponent $c$ (panel e) also indicates the boundary. In the lower layer, $c$ is close to 1, corresponding to a Debye relaxation. Compared to the other parameters, less literature is available for the relaxation exponent and the values are close to each other. Therefore, $c$ seems less suitable for an interpretation in terms of material properties. The high frequency value $\varepsilon_{HF}$ (panel c) is the only one which does not show a clear distribution. The range in this case is significantly smaller compared to the other parameters, which could make it more difficult to identify differences in materials based on $\varepsilon_{HF}$. This

is the case in this example but could be different for other test sites or under different condition. The region of slightly higher values at about two meter depth on the second half of the profile could indicate a systematic change in the permittivity.

The two layers could be identified as the snow layer and the underlying bedrock, expected as limestone layer described by Rowan (1993), which could also be seen at some spots on the surface near the profile area. The resistivity and the relaxation time on the first half of the profile show some more variation underneath the snow. This could be caused by a third layer of

weathered material on top of the bedrock or a possible ice cover underneath the snow, as described by Scherler et al. (2010).

The determined values in their dominating range for the two horizontal regions of profile B-SCH are compared to literature in table 1. The literature values of the different materials can vary over large ranges, which is mainly caused by the differences in physical conditions. For ice and snow the purity, density, salinity and temperature can strongly influence the electrical parameters (Auty and Cole, 1952; Arenson et al., 2015; Evans, 1965). New and soft snow, as it was present in case of the

Profile B-SCH, has a relatively low density, meaning an increase of resistivity towards more dense snow. For the literature values of ice the attribute pure means that there is no impurity caused by other materials in the ice/water, but there are still variations based on the physical conditions like temperature. If ground material is frozen, like on permafrost or seasonal frost, the measured signal is expected to react as a composition of the basic material and ice (Zorin and Ageev, 2017). The parameters for frozen ground are strongly dependent on the ice content and temperature (Arenson et al., 2015; Grimm et al., 2015). In the

case of limestone, laboratory investigations of CCR by Murton et al. (2016) showed, that the resistivity of a frozen limestone sample ($10^5\,\Omega\mathrm{m}$) can be one order of magnitude higher than for the same sample in an unfrozen state.

The electrical behaviour of water needs particular attention, because the typical relaxation frequency of water is in the range of GHz (e.g. Artemov and Volkov, 2014). Consequently, the high frequency value $\varepsilon_{HF}$ of about 5 will not be reached within our frequency range. Instead, the real dielectric permittivity exhibits a constant behaviour, denoted as the low frequency value,

which is commonly known as 80. However, a study by Seshadri et al. (2008) shows, that even in water without solids another relaxation process takes place at lower frequencies in the range of Hz, associated with interfacial polarization due to ions. The dielectric permittivity therefore can reach values of more than 1000 at the lower boundary of our frequency range, especially if the water has a high electrolyte concentration.

In table 1 (top row) the information of the inversion results (Fig. 7) is extracted and attributed to different layers. For the first

layer, the estimated values agree for all Cole-Cole parameters with those of snow. For the area under the horizontal boundary,

**Table 1.** Results of the 2-D inversions on profile B-SCH and the profile on Lake Prestvannet and comparison with literature values for snow, ice, water and limestone, which is expected as bedrock material on B-SCH. The characteristic parameters from the inversions are given as the range for the two horizontal layers for profile B-SCH and the two vertical separated regions of lake and shore for the profile on the Lake Prestvannet. Literature values were taken from Arenson et al. (2015)[1], Evans (1965)[2], Achammer and Denoth (1994)[3], Palacky (1988)[4], Murton et al. (2016)[5], Olatinsu et al. (2013)[6], Seshadri et al. (2008)[7], Artemov and Volkov (2014)[8] and Auty and Cole (1952)[9].

| | | $\rho_{DC}[\Omega m]$ | $\varepsilon_{DC}[-]$ | $\varepsilon_{HF}[-]$ | $\tau[s]$ |
|---|---|---|---|---|---|
| B-SCH | First Layer | $10^6$ - $10^7$ | 15 - 100 | 2 - 9 | $10^{-5}$ - $10^{-4}$ |
| | Second Layer | $10^3$ - $10^6$ | 50 - 700 | | $10^{-4}$ - $5 \cdot 10^{-4}$ |
| Lake Prestvannet | Lake | $10^4$ - $2 \cdot 10^4$ | 750 - 3000 | 5 - 11 | $10^{-4}$ - $6 \cdot 10^{-4}$ |
| | Shore | $4 \cdot 10^5$ - $10^4$ | 850 - 6500 | 12 - 15 | $6 \cdot 10^{-5}$ - $2 \cdot 10^{-4}$ |
| Literature | Snow$_{1,2,3}$ | $10^5$ - $10^8$ | $\sim 40$ | $< 15$ | $\sim 10^{-4}$ |
| | Limestone (unfrozen/frozen)$_{4,5,6}$ | $10^3$ - $10^5$ | 50 - 130 | 5 - 9 | $2 \cdot 10^{-5}$ - $> 10^{-4}$ |
| | Ice (pure)$_{2,7,8,9}$ | $10^5$ - $10^9$ | 92 - 105 | 3 | $2.2 \cdot 10^{-5}$ - $5 \cdot 10^{-3}$ |
| | Water $_{2,7,8}$ | $< 10^6$ | 80 | $5 (> GHz)$ | $10^{-12}$ - $10^{-10}$ |

indicated by the clear change in parameters $\rho_{DC}$, $\varepsilon_{DC}$ and $\tau$, the estimated values are in agreement with the expected limestone layer. As the resistivity indicates a further separation, the area directly underneath the snow could either belong to an ice layer or be caused by a frozen state, followed by a less resistive part of the limestone. The observation that all other parameters except resistivity show no significant change in this area can be explained by the similarity of the parameters for ice and limestone, indicated by the literature values. The similarity and small range of high frequency permittivity $\varepsilon_{HF}$ of all corresponding literature values can explain the low variability and missing distinctness in this parameter section. In summary, the determined Cole-Cole parameters from the 2-D inversion are consistent with the literature values for the expected materials.

A comparison with the results of the single site inversion illustrates the differences of the inversion methods. Figure 8 shows the results for the same measurements as in Fig. 7 but evaluated by the single site inversion. The parameters are determined individually for each quadrupolar measurement. In order to represent the results in a two-dimensional structure, the parameters are assigned to a certain location according to the midpoint position and extent of the array. Thus, a two-dimensional pseudo-section is finally created for each parameter, which can provide a rough overview of the subsurface structure. The difference between Fig. 7 and Fig. 8 is that Fig. 8 represents an inversion result only with respect to frequency, but a pseudo-section with respect to the spatial distribution, where as Fig. 7 is a full inversion result with respect to both frequency and space. As expected when comparing pseudo-sections with 2-D inversion results, overall structures are similar, but there are differences in detail. The low-frequency permittivity (panel b) and the relaxation time (panel d) systematically increase with depth, while the resistivity (panel a) shows a decrease with depth, all indicating a horizontally layered structure. As seen before in the 2-D results, the high permittivity value $\varepsilon_{HF}$ exhibits a small range of values and does not show a systematic distribution, but fits in the range of the literature. Since the results of the deeper pseudo-layers always represent an integral value over the entire

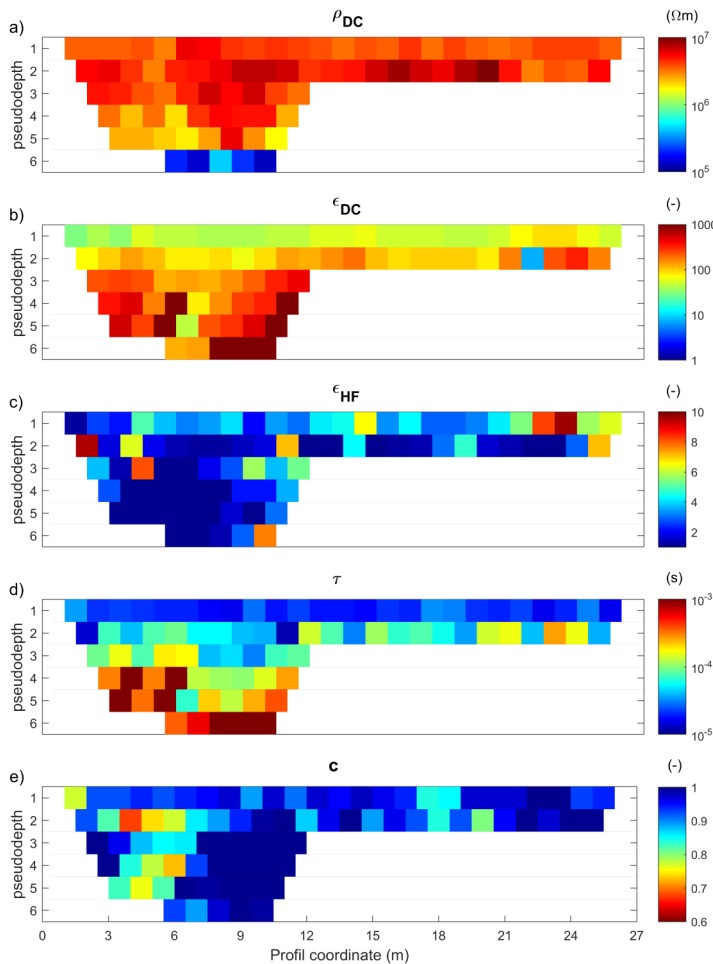

**Figure 8.** Pseudosections of the five Cole-Cole parameters (a-e) calculated by the single site inversion for profile B-SCH. Measurements were done for larger configurations on the left half of the profile.

depth range, it has to be expected that clear boundaries between the layers can not be identified by this method. Therefore the measurements have to be analysed in dependence to each other, as it was done by the 2-D inversion. However, since the inversion with respect to the frequency dependence is independent for the two methods, we take the qualitative consistency as evidence that our 2-D inversion is a feasible tool to invert spatial and frequency dependence at the same time.

## 4.2 Tromsø

The investigations at Lake Prestvannet focus on the vertical transition from the lake to the shore rather than on the distribution with depth. In Fig. 9, the result of the AarhusInv 2-D inversion is shown for all Cole-Cole parameters. Measurements were carried out for just a small spacing (Wenner $a = 1.5\,\mathrm{m}$), so the penetration depth is not more than approximately $1.5$ meters. Brighter areas are again below the DOI and the vertical black line indicates the surface position of the transition from the lake to the shore at profile meter $20.5$. Except for the relaxation exponent $c$ (panel e), in all models the transition is defined by a parameter change. The ranges of the estimated Cole-Cole parameters for lake and shore are given in table 1 and can be compared to the literature values of snow, ice and water.

The high-frequency permittivity $\varepsilon_{HF}$ (panel c) shows lower values for the lake than onshore. This parameter is somewhat specific, because the variation from lake to shore is the only prominent variation, whereas the other parameters also show some structure within both sides. Compared to profile B-SCH (Fig. 7), where $\varepsilon_{HF}$ has a small dynamic range with little spatial coherence, for the data of Lake Prestvannet both permittivity values $\varepsilon_{DC}$ and $\varepsilon_{HF}$ seem to be useful to distinguish different materials or the state of freezing. The low-frequency permittivity $\varepsilon_{DC}$ (panel b) shows the transition from lake to shore, but exhibits additional variation on either side of the transition. The land side shows an anomaly close to the transition. The cause is not exactly known, but we hypothesize that it indicates a change in sediments. A detailed analysis, where the two properties of permittivity may be combined with resistivity in a multi-parameter analysis may be a subject of future research. The relaxation time $\tau$ shows relatively homogeneous values for each of the sides. The fact, that the relaxation times of snow are shorter than those of ice (Evans, 1965) is consistent with our results. The resistivity $\rho_{DC}$ decreases from higher values on the snow covered land side, by about one order of magnitude on the lake. The relaxation exponent $c$ shows small variations and poor spatial coherence, and is difficult to interpret in terms of material variations.

Onshore, the measurements were taken on the snow and the known values of snow are consistent for some parameters. Higher density of the snow could explain the much lower resistivity than obtained for the snow at B-SCH. However, in combination with the values of $\varepsilon_{DC}$ which are higher than expected for snow, this could indicate that the measurements are under the influence of the ground material beneath. For the frozen lake, the values estimated from the inversion are a bit higher than typical literature values for ice. However, it is known that the electrical parameters of frozen water bodies can be very different from those of pure ice. The lake ice could be a composition of ice and partly water, instead of a pure ice body. Such mixtures can result in higher values of permittivity. The high salinity of the lake can further increase the low frequency permittivity of the water to the range of $> 1000$ (Seshadri et al., 2008). This is significantly larger than the value of pure water of about $80$ (Evans, 1965), and could explain the high $\varepsilon_{DC}$ values determined by the inversion. At the high-frequency end in our frequency range, the value of $\varepsilon_{HF}$ is probably also controlled by water (around 80 in that frequency range), which explains why our estimated value is larger than that expected for pure ice. Lower values of resistivity could as well be explained by this composition because water has a lower intrinsic resistivity than ice. A similar observation was made by Przyklenk et al. (2016) during their discussion of measurements on mountain ice.

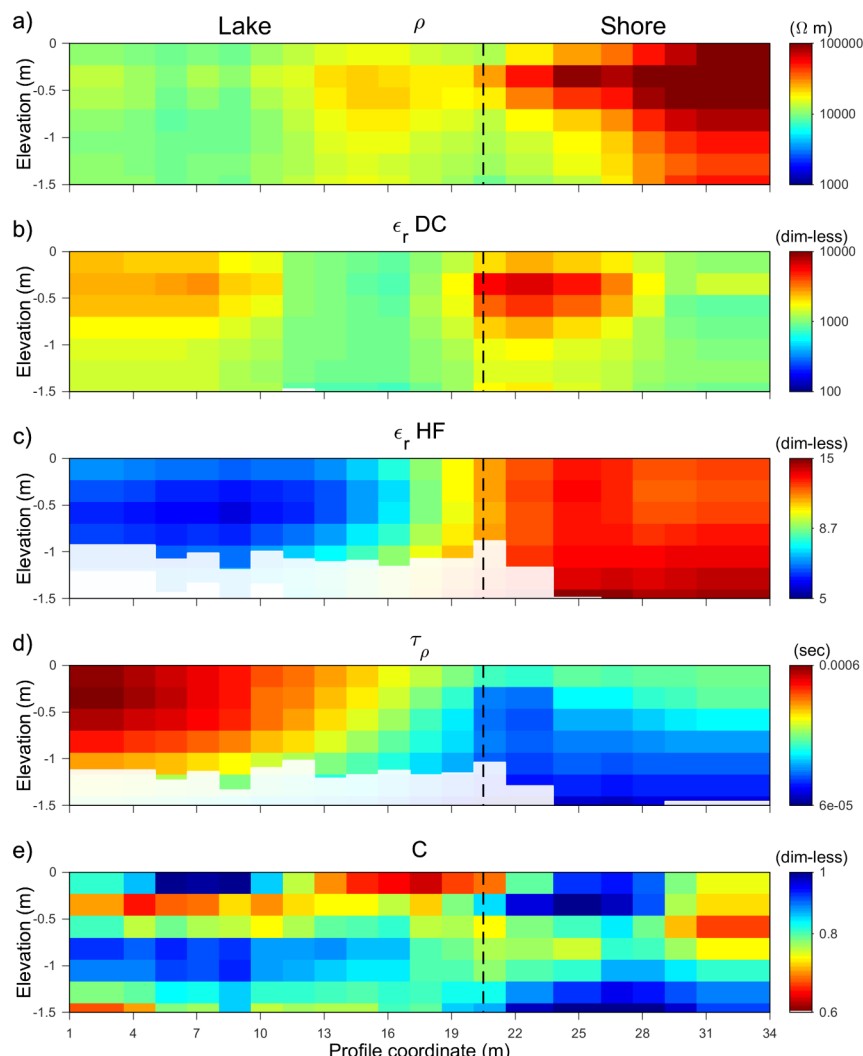

**Figure 9.** 2-D inversion result of the 5 Cole-Cole parameters (a-e) for the measurements at the lake Prestvannet denoted in Fig. 3. The surface boundary between lake and shore is at profile meter 20.5, indicated by the dashed line, where a change in most parameters can be recognized. The whitened areas are those where the value of DOI is exceeded.

## 4.3 Data fit

For the assessment of the inversion results, we consider their quality in terms of the data fit. Because of being a spectral inversion, the inversion includes the fit of all measured data over frequency, corresponding to a spectral pair of magnitude and phase shift for every 4-point-array. In the following, the data misfit is expressed in terms of the weighted mean square error, where each difference is weighted by the inverse of the data error. This value is denoted by the symbol $\chi$, and is a well

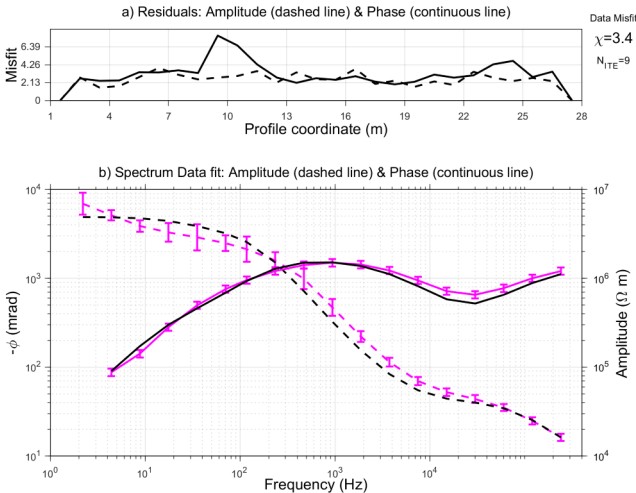

**Figure 10.** Data fit of the 2-D inversion of profile B-SCH. The top panel shows the misfit of amplitude (dashed line) and phase shift (solid line) over the profile. The bottom panel shows an example of measured (red) and inverted (black) spectra of amplitude (dashed lines) and phase (continuous lines), which belong to the dipole-dipole measurement starting from profile meter 16 ($a = 1\,\mathrm{m}$, $n = 1$) along the profile direction (see fig. 7). The measured data is the same as in figure 4, with resistivity instead of impedance, and phase shift on a logarithmic scale. The total data misfit is $\chi = 3.4$ after 9 iterations. For each profile coordinate, the misfit of all data corresponding to this point were averaged to obtain the top panel.

established measure of the misfit. Together with an additive regularization term, it is minimized during the inversion (Fiandaca et al., 2013).

Figure 10 shows the data fit for the 2-D inversion of the Schilthorn measurements. The top panel (a) shows the misfit of amplitude and phase shift over the profile length. The residuals are averaged over all measurements with all configurations

with the same midpoint. The total inversion misfit is $\chi = 3.4$, based on the average relative standard deviation of amplitude (0.16) and phase measurements (0.10). The inversion converged after nine iterations. The misfits of amplitude and phase are homogeneous over the profile, except around profile meter 10, where the phase misfit is significantly higher. As can be seen from Fig. 7, this is the area where $\rho$ and $\tau$ show another change for the deeper region. The higher misfit corresponds to the data of larger configurations (dipole-dipole with $n = 5, 6$), where the measured signals show slightly different curves than for

the shallower measurements. The large misfit indicates that the deep structure should be treated with caution because it could be caused by difficulties in matching data.

Panel (b) of Fig. 10 shows the data fit of the spectrum, which was previously shown during the discussion of the single site inversion (Fig. 4). Data and inversion results are shown for the amplitude (dashed lines) and the phase (continuous lines). The amplitude is not exactly the same as the magnitude in Fig. 4, but was converted to the frequency dependent resistivity

(using Eq. (3)). The negative phase shift is displayed on a logarithmic scale in $mrad$. Some data points, in this case for the two lowest frequencies, that caused difficulties for the inversion code and were identified as outliers, are not shown. Overall,

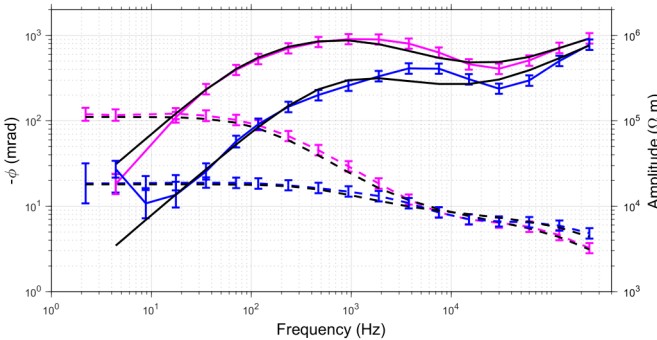

**Figure 11.** Data fit of the 2-D inversion for two stations of the lake Prestvannet profile. The blue lines correspond to a measurement at $11.5\,\mathrm{m}$, which is on the lake, the red lines to an onshore measurement at $28\,\mathrm{m}$, with the corresponding inverted spectra (black). Both were measured with a Wenner configuration ($a = 1.5\,\mathrm{m}$). The measured data are the same as in figure 5. The amplitude is indicated by the dashed lines and the phase shift by the solid lines.

for both amplitude and phase shift, the shape of the spectrum is well matched. For several data points, the calculated curve is not within the data errors, which is reflected in a misfit value $\chi > 1$. The data errors are calculated by the device by stacking multiple measurements. Considering that broadband electrical data of 79 spectra were matched with a single 2-D model, we find the fit satisfactory. The 19 discrete measured frequencies seem to be sufficient to define the dispersion of permittivity.

Another example of data spectra is shown in Fig. 11 for the measurements taken at Lake Prestvannet. The data are the same as discussed in Fig. 5, with one lake measurement (blue) and one onshore measurement (red) corresponding to profile coordinates $11.5\,\mathrm{m}$ and $28\,\mathrm{m}$ from Fig. 9. The total data misfit of the inversion is $\chi = 1.8$. The onshore spectra shows some similarity to the Schilthorn spectra (Fig. 10). The amplitude and the phase shift are both matched very well. As mentioned before for the single site inversion, the spectra of lake and land show a different frequency-dependent behaviour. A slight difference between

the measured and calculated data is visible for the phase of the lake measurement in the intermediate frequency range. This could indicate limitations of the single Cole-Cole model.

  The sensor height effect is not negligible in this case. The lowest two frequencies seem to be affected and can not be matched by the inversion. This is the same effect as discussed previously during the single site inversion.

## 5 Conclusions

Wide-band complex resistivity measurements based on capacitively coupled electrodes were carried out on two cryospheric field sites, on a frozen lake in Norway and in an alpine region in Switzerland. By recording the spectral data in an intermediate frequency range, where conduction currents and displacement mechanisms are relevant, the determination of the frequency-dependent electrical resistivity and permittivity is investigated. The data analysis is done by a novel 2-D inversion for broad-band electrical measurements based on the inversion tool AarhusInv, where the permittivity is parameterized with a

Cole-Cole model.

The first applications of the 2-D inversion give encouraging results in the sense of consistence with known materials and structure. For our shallow field measurements, the single Cole-Cole model seems sufficient and there is no evidence of fundamental difficulties in fitting spectral data. The observed misfits are acceptable in a sense that $\chi$ is close to 1, in the range typical for conventional 2-D resistivity inversions, and should have similar causes, such as 3-D effects. In principle, it is possible to implement a double Cole-Cole model, which could fit more complex spectra, but has more inversion parameters. The assumption of zero sensor height seems to be uncritical in our chosen field applications. In some cases, it could be helpful to discard some low-frequency data, which are most strongly affected by electrode height above the observed surface.

The determination of the electrical parameters for both investigations was successful. They show reasonable consistence with literature values within a maximum deviation of one order of magnitude. The inversion for the five Cole-Cole parameters works as well as conventional 2-D resistivity inversion, except for the frequency exponent, which tends to show spatially incoherent images. The complementary information provided by the high- and low-frequency limits of permittivity can be significant. Some structures are more clearly defined than in the corresponding resistivity image. We conclude that using different parameter sections for the interpretation can lead to a more differentiated analysis of the subsurface.

The full spectral information can be used for the determination of ground ice content at the field scale, as suggested by Grimm and Stillman (2015). This is an objective of research in periglacial environments, and will be a subject of future work.

*Acknowledgements.* We are grateful to Katharina Bairlein (PTB, Braunschweig) and Christian Kulüke (TU Braunschweig) for the support of our measurements in Norway and Switzerland. We thank Wim Weber (City of Tromsø) for the permission to take the data on the lake Prestvannet.

The work was sponsored by the German Research Foundation (projects HO 1506/22-1 and HO 1506/22-2), and by the University of Aarhus.

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
