# Peer review of "Two-dimensional Inversion of wideband spectral data from the Capacitively Coupled Resistivity method - First Applications in periglacial environments"

_The Cryosphere, 2018_

## Referee Comment (RC1) · Anonymous Referee #1 · 4 Apr 2019

Two-dimensional inversion of wideband spectral data from the capacitively coupled resistivity method – First application in the periglacial environment.

By Mudler et al.,

Major

The manuscript describes the new development in the capacitively coupled method which was developed for low-frequency measurements. Authors modified the method by including Cole to Cole parameterization. However, for someone not deeply familiar

with these methods it hard to follow the manuscript. It was very unclear from reading an abstract: Why is it important to modify low frequency method in first place? What do low frequency methods provide? Why it is important to extend them to wider frequency range? Which additional rate of frequencies does this new modification cover? What type of information do we get by inverting CCR data?

Similarly, in the introduction, authors jump on explaining how by having electrical resistivity and dielectric permittivity is not enough. I suggest to start with explaining why doing ERT measurements is important in the first place? What type of subsurface information do we obtain by using these ERT? Then move to explaining why it is not enough and that having permittivity provides and additional information that is useful for interpretation of subsurface conditions. It is not clear, which subsurface conditions authors are referring to?

First paragraph ends with statement that determination of the ice content is possible with ERT. Is that the overall goal of this work?

P2L10 Why is it usable on the extremely hard surface? Need to better explain it. From the description, I not sure what type subsurface information CCR provides.

P2L30. OK, the aim of the study is test the application of the newly developed method on the identification of the ground ice.

Shilthorn From the description of the subsurface I conclude that it is a rock. What type of ground ice can exist in the solid rock? Does that rock has fractures that filled with ice? What about ice that might be formed at the ground surface? Does that ice layer is important and was taken into account?

Lake site Lake was frozen. Is there any information of the ground subsurface? Is it frozen? How deep is the seasonal frost layer? Any information on the percentage of ice within the ground?

P6. L10. There some GRP measurements in permafrost regions that estimate ALT

and soil moisture, and could be used to calculate ice content (e.g. Chen et al., 2016 and Jafarov et al., 2017).

P6.L15 What is relatively high? Do you mean ice lenses wise or massive ice?

P6.L15-22 lit review and can be moved to the introduction.

P7.L25-30 Does that mean that inversion depends on one parameter (c)?

P8.L23. Figure 4 inversion done with and without determination of height. Where are those two on the plot? I do not see two curves (one for h0 another for hinv)? The legend should be adjusted correspondingly. X-axis, is f an actual frequency or logarithm?

Figure 6. It is not clear which of the Tromso data correspond to the lake ice and which to the ground ice?

P12.L16 Why authors decided to use AarhusInv code and not BERT for example? How well does AarhusInv compares to other existing codes? Is this code an open source? If it is, then it would nice to provide a link for the modification implemented in the code.

Why did authors choose $\chi$ metric? Is that commonly acceptable fitness metric? Why not RMSE or Taylor diagram?

P20.L17 'reasonably consistent' ... Is that possible to quantify it (what is the correlation)?

Overall, I have been struggling throughout this paper to understand the purpose of this study. What is an ultimate goal of doing this? Is it to get a better measurement of the ground ice? If yes. Are there any ground truth data? How these inversion can be compared with in-situ data?

Suggestions: In this current version of the manuscript, methods, results, and literature review are all mixed up together. Think how you can better organize/separate them. Starting from the bigger picture, like knowing ground ice is extremely important for many reasons... In particular, for better understanding of the permafrost thawing rates

and consequences. Then introduce the method. Provide a literature review on the existing methods and models. Justify the usage of the current model and talk about how important the current improvements are in terms of better quantifying of the ground ice. In addition, in the description of the site location, it would be extremely useful to know subsurface characteristics/properties. Are there fractures in the rock? How much do you know about subsurface ground ice at the lake station? Comparing inversely derived ground ice with actual ground ice will be extremely useful.

The current version is a good methodological paper and missing emphasis on how this work is important and how it is contributing the current state of science. Addressing these two missing issues will make this paper suitable for the journal like Cryosphere.

References

Chen, A., Parsekian A., Schaefer K., Jafarov E., Panda S., Liu L., Zhang T., and Zebker H: 2016. Ground-penetrating radar-derived measurements of active-layer thickness on the landscape scale with sparse calibration at Toolik and Happy Valley, Alaska. GEOPHYSICS, 81(2), H1-H11. doi: 10.1190/geo2015-0124.1

Jafarov, E. E., Parsekian, A. D., Schaefer, K., Liu, L., Chen, A. C., Panda, S. K. and Zhang, T. (2017), Estimating active layer thickness and volumetric water content from ground penetrating radar measurements in Barrow, Alaska. Geosci. Data J.. doi:10.1002/gdj3.49

---

## Referee Comment (RC2) · Anonymous Referee #2 · 2 May 2019

**REVIEW**
tc-2018-288
Two-dimensional Inversion of wideband spectral data from the Capacitively Coupled
Resistivity method - First Applications in periglacial environments
Mudler et al.

**SUMMARY**

This paper presents data and modelling results for broadband spectral capacitive
resistivity experiments performed in cold regions. The experiments appear sound in
design, and the data are novel and very interesting from the perspective of
electrical/electromagnetic geophysics and cold-regions research. However, the paper has
a few shortcomings. The objectives of the paper are not entirely clear at first. Is the focus
on SIP or CR? It becomes clear (I think) that the focus is on cold-regions application of
broadband spectral CR. If this is the desired focus, the paper would be made more
impactful by including: 1) a review of electrical, IP, SIP, and CR applications to cold
regions and permafrost; 2) a clear description of the benefits, limitations, and favourable
conditions for broadband spectral CCR, and how these relate to cold regions; and 3) a
more thorough analysis of the inversion results in terms of cold-regions ground properties
of interest such as water content, ice content and temperature.

**DETAILS**

Abstract: Lacks focus on objective and results.

p1,L19: Why? High conductivity material exhibit spectral characteristics as well.

p2,L28: What about Routh et al. (1998), Kemna et al. (2000) and several works
thereafter?

p.2,L30: Introduction is somewhat unclear. It starts out focussed on SIP, then CCR, but
then states the objective as investigating the field applicability of [spectral] CCR in cold
regions. To support the latter, the intro needs a little more background on electrical
geophysics and material properties in permafrost and/or glaciology.

eq.1: Although somewhat semantic, I view the low-frequency CCR experiment as
responding to the complex conductivity where the imaginary conductivity has a
contribution from the real dielectric. Of course, at higher frequency and in the presence of
ice, permittivity may be more relevant. However, I do not think that equation 1 should be
referred to as representing the "complex permittivity." Consider "effective permittivity"
which has a contribution from the imaginary conductivity. The distinction is important
because the true dielectric permittivity is what results in wave propagation in Maxwell's
equations, not the imaginary conductivity. Furthermore, to talk only of displacement
currents denies the possibility of IP-type currents which may dominate at lower
frequency.

p.6,L4: You say the "conduction current" is in-phase, but then you say that IP is concerned with the conduction current part of the impedance. You need to be clearer on the distinction between the real conductivity (conduction current), the imaginary conductivity (IP current), and the real permittivity (displacement current).

eq.4: Again, consider noting that any IP effect will be wrapped up in here as $\varepsilon_{eff} = \varepsilon_R + \sigma_I/\omega$.

p.7,L8: The height effect is also discussed by Wang et al. (2016) but in a shady pseudo-journal. The authors could decide if it warrants consideration.

p.8,L3: You say the inversion is frequency dependent, but then go on to say that the system response is controlled by geometry, not frequency. Clarify.

p.8,L5: You need to thoroughly describe the "operating range" of CCR with respect to treatment of the data for the single site inversion and the 2D inversion. Application of a 2D resistivity inversion (with geometry-based sensitivity) requires low-induction number conditions (which actually appears to be violated for some of your lower frequency-resistivity combinations). Does the single-site inversion require LIN conditions? What about wave effects? For some of the high frequency-resistivity pairs encountered, quasi-static conditions are violated and a true permittivity will result in wave propagation. This should not(?) affect CC model fits, but it should(?) affect the 2D inversions using a resistivity-type sensitivity function.

[Figure]

a = 1.5 m, $\varepsilon_R = 3$

p8,L6: "Induction effects" would typically be understood to mean inductive source effects of current-carrying cables. You don't have these. So, do you mean magnetic coupling as described by McNeil?

p.13,L25: Well, the dielectric constants for rock and snow are both around 3-5, so...

p.13,L32: Snow cover typically inhibits frozen ground.

Table 1: Add water. More discussion is required in comparing recovered values to expected material properties.

p.15,L5: In comparing to literature values, what about the observations by Weigand and Kemna (2016) that SIP model parameters obtained from a CC model are biased? Is this alleviated by having c as a free parameter and/or by having 19 points in the spectrum?

Fig.8: Use same scales as Fig.7. Are some of the observed differences attributable to height effects or breakdown of LIN conditions?

p.18,L5: Why is the DC permittivity so (unreasonably) high?

p.18,L11: Is there any benefit to setting c constant (i.e., choosing a decomposition for the CC model). Is it reasonable for c to show so much variability? Is it highly sensitive, and if so, is it just absorbing error in the inversion?

p.18,L19: Actually, the LF permittivity of water is around 80, but you need to get up to $10^{10}$ to $10^{12}$ Hz before it drops to around 3.

Figure 10: a) Use dash and solid. b) What is the distinction between black and purple lines?

**REFERENCES**

Kemna A., Binley A., Ramirez A. and Daily W. 2000. Complex resistivity tomography for environmental applications. Chemical Engineering Journal 77, 11–18.

Routh P.S., Oldenburg D.W. and Li Y. 1998. Regularized inversion of spectral IP parameters from complex resistivity data. Expanded Abstracts of the 68th Annual International Meeting, Society of Exploration Geophysicists, 810–813.

Wang et al., 2016. Investigation on a Novel Capacitive Electrode for Geophysical Surveys. Journal of Sensors, 4209850.

Weigand M. and Kemna A. 2016. Relationship between Cole-Cole model parameters and spectral decomposition parameters derived from SIP data. Geophysical Journal International 205, 1414–1419.

---

## Referee Comment (RC3) · Anonymous Referee #2 · 2 May 2019

Axis labels are switched on figure for Review 2. Resistivity is on the y-axis and frequency on the x-axis.

---

## Author Comment (AC1) · 14 Jun 2019

**Review 1 (04.04.2019)**

**R**: Referee's comment

**A**: Author's response

**C**: Change in manuscript

**(1)**
**R:** The manuscript describes the new development in the capacitively coupled method
which was developed for low-frequency measurements. Authors modified the method
by including Cole to Cole parameterization. However, for someone not deeply familiar with these
methods it hard to follow the manuscript. It was very unclear from reading
an abstract: Why is it important to modify low frequency method in first place? What do
low frequency methods provide? Why it is important to extend them to wider frequency
range? Which additional rate of frequencies does this new modification cover? What
type of information do we get by inverting CCR data?

**A:** We can measure with our method from 1Hz up to 240 kHz, IP works usually in the range of mHz up
to 1kHz. The higher frequencies in combination with the high resistivity conditions enable to
estimate the resistivity as well as the permittivity of the subsurface. Ice (and snow) has a
characteristic relaxation process within our frequency range at around 10kHz, which is what makes
the extension in frequency range so important. The Cole-Cole model gives a possibility to fit the
spectra of these data. We can determine the 5 Cole-Cole parameters and thereby characterize the
subsurface. The modification of the inversion algorithm is necessary because compared to existing IP
inversions we need one more parameter.

**C:** The abstract will be rewritten such that the aim of the study and the issues mentioned above
become clearer. Also, sections of the introduction will be rewritten to describe better what
information can be provided by other methods, and what is new here what we try to obtain out from
our method.

**(2)**
**R:** Similarly, in the introduction, authors jump on explaining how by having electrical resistivity and
dielectric permittivity is not enough. I suggest to start with explaining why
doing ERT measurements is important in the first place? What type of subsurface information do we
obtain by using these ERT? Then move to explaining why it is not enough and that having permittivity
provides and additional information that is useful for interpretation of subsurface conditions. It is not
clear, which subsurface conditions authors are referring to?

**A:** We agree with the idea of starting from the ERT and then explain the benefits of our method. The
subsurface conditions we are focusing on are periglacial areas. The main reason is that the method is
sensitive to the presence of water ice, due to the characteristic frequency dependence of electrical
permittivity.
However, in this paper we focus on methodological aspects, such as the questions: Is it possible to
measure the CCR with our unique equipment in areas of (possible) occurrence of ice/permafrost?
How strongly are the data affected by the electrode height? Can we invert our data with the new
spectral 2D inversion and are results consistent with the subsurface structure and literature values?

The quantitative estimation of ice content is an ultimate goal of future developments, for which we lay the foundation with this work.

**C:** We will rewrite several sections of the introduction. First we focus more on the existing studies and methods, starting from ERT and then explaining the role of spectral induced polarization (SIP). We will explain the characteristic relaxation of ice and make clear why our measurement uses these aspects to obtain specific additional information.

**(3)**
**R:** First paragraph ends with statement that determination of the ice content is possible with ERT. Is that the overall goal of this work?

**A:** This might be subject of future work. The statement was intended to point out what could be possible (and was already discussed by several authors). It is not the focus of this work.
**C:** The focus of the work will be explained more clearly. The introduction will be changed accordingly (see (2)).

**(4)**
**R:** P2L10 Why is it usable on the extremely hard surface? Need to better explain it. From the description, I not sure what type subsurface information CCR provides.

**A:** On hard surface the capacitive coupling gives an advantage over the "normal" coupling with skewers (no drilling or watering of electrodes needed). This is explained in P2 L10-13.
The information are resistivity, permittivity and the characteristic relaxation information given by the other CC parameters (tau, low-frequency permittivity).
**C:** see (1) (2)

**(5)**
**R:** P2L30. OK, the aim of the study is test the application of the newly developed method on the identification of the ground ice.

**A:** Yes.
**C:** See above. We will try to make this clear from the beginning.

**(6)**
**R:** Shilthorn From the description of the subsurface I conclude that it is a rock. What type of ground ice can exist in the solid rock? Does that rock has fractures that filled with ice? What about ice that might be formed at the ground surface? Does that ice layer is important and was taken into account?

**A:** The surface under the snow is a layer of limestone (described in the text). Unfortunately, we do not know details, like if its fractured. We do not have a high investigation depth, so just Ice in very shallow depth would influence the measurements. But main focus lies on the structural aspects.
**C:** Will formulate this more precisely and add information.

**(7)**
**R:** Lake site Lake was frozen. Is there any information of the ground subsurface? Is it frozen? How deep is the seasonal frost layer? Any information on the percentage of ice within the ground?

**A:** Unfortunately, there is no further information of the ground subsurface. The measurements were made with the idea to test whether the transition between lake and land is visible in our data. A correlation with geology, or quantitative estimation of ice content, was not a primary goal. Therefore, we can make an assessment of the data only in a qualitative sense.

**C:** We will modify the corresponding sections such that it is clear from the beginning that we are not aiming at a quantitative correlation with ground properties, but that we consider the qualitative assessment sufficient at this stage.

**(8)**
**R:** P6. L10. There some GRP measurements in permafrost regions that estimate ALT (active layer thickness) and soil moisture, and could be used to calculate ice content (e.g. Chen et al., 2016 and Jafarov et al., 2017).

**A:** The role and possibilities of GPR will discussed a little more.
**C:** This will be included in the text.

**(9)**
**R:** P6.L15 What is relatively high? Do you mean ice lenses wise or massive ice?

**A:** This statement just means that the existence of ground ice, as in periglacial areas, leads to high resistivity without the distinction between massive or ice lenses. The "relatively" might be misleading.
**C:** The sentence will be formulated more concrete.

**(10)**
**R:** P6.L15-22 lit review and can be moved to the introduction.

**A:** We agree
**C:** Will be included in the introduction (see (2)).

**(11)**
**R:** P7.L25-30 Does that mean that inversion depends on one parameter (c)?

**A:** No, that is not the case. We just want to say that the other 4 CC-Parameters can directly be related to a physical context and there exist material-specific literature values. There are values for c as well, but just rare.
**C:** Sentence P7 L28 f. will be changed and formulated more precisely.

**(12)**
**R:** P8.L23. Figure 4 inversion done with and without determination of height. Where are those two on the plot? I do not see two curves (one for h0 another for hinv)? The legend should be adjusted correspondingly. X-axis, is f an actual frequency or logarithm?
Figure 6. It is not clear which of the Tromso data correspond to the lake ice and which to the ground ice?

**A:** As written in the figure caption, the two variants of inversion cannot be distinguished from each other and therefore are shown as one line (see legend).

The x-axis is the actual frequency, but shown in logarithmic steps. Axis and the corresponding label are standard, so we are not sure what causes the confusion. At this stage, in fig. 6 there is no distinction between lake site and land site.

**C:** Fig.4 (same Fig. 5c and d) will be changed that for h_0 and h_inv separate curves will be shown, but they will lie on each other.

Fig. 6 will be changed, such that there is a distinction between lake site and land site at Tromso data by using two different colors. This will be explained in the text (P11 L7 ff.).

Concerning the frequency axis, we do not see how to change anything, as we follow common standards.

**(13)**
**R:** P12.L16 Why authors decided to use AarhusInv code and not BERT for example? How well does AarhusInv compares to other existing codes? Is this code an open source?
If it is, then it would nice to provide a link for the modification implemented in the code.
Why did authors choose $\_(CHI-)$metric? Is that commonly acceptable fitness metric? Why not RMSE or Taylor diagram?

**A:** Because two of the authors are developers of AarhusInv, working at Aarhus University. Using BERT would cause the same procedure of extending the code for our aims.
AarhusInv is a freeware for non-commercial purposes (http://hgg.au.dk/software/aarhusinv/). The code is not open source. The complex impedance is modelled in 2-D solving the Poisson's equation, Fourier transformed in the strike direction, without considering EM effects (Fiandaca et al., 2013), as done for instance in the complex resistivity code cR2 developed by Andrew Binley (http://www.es.lancs.ac.uk/people/amb/Freeware/cR2/cR2.htm).
The data misfit values are expressed in terms of chi values, because the objecting function minimized in the inversion process is the sum of the data and regularization chi values (Fiandaca et al., 2013).

**C:** We will add this information about AarhusInv this in chapter 4.

**(14)**
**R:** P20.L17 'reasonably consistent' … Is that possible to quantify it (what is the correlation)?

**A:** Very difficult to quantify because we do not have exact values for resistivity and permittivity from other methods or even better from laboratory analysis. This statement should mean that the determined values fit in the range of literature values of what we know and what we expect from the subsurface at the test sites.
**C:** Will be formulated more precisely.

**(15)**
**R:** Overall, I have been struggling throughout this paper to understand the purpose of this study. What is an ultimate goal of doing this? Is it to get a better measurement of the ground ice? If yes. Are there any ground truth data? How these inversion can be compared with in-situ data? Suggestions: In this current version of the manuscript, methods, results, and literature review are all mixed up together. Think how you can better organize/separate them. Starting from the bigger picture, like knowing ground ice is extremely important for many reasons... In particular, for better understanding of the permafrost thawing rates and consequences. Then introduce the method. Provide a literature review on the existing methods and models. Justify the usage of the current model and talk about how important the current improvements are in terms of better quantifying of the ground ice. In addition, in the description of the site location, it would be extremely useful to know subsurface characteristics/properties. Are there fractures in the rock? How much do you know about subsurface

ground ice at the lake station? Comparing inversely derived ground ice with actual ground ice will be extremely useful. The current version is a good methodological paper and missing emphasis on how this work is important and how it is contributing the current state of science. Addressing these two missing issues will make this paper suitable for the journal like Cryosphere.

**A:** We appreciate to positive evaluation as a good methodological paper, and we are thankful for the constructive comments. We will try to address the issues and hope that we can bring the paper into a suitable form.

The purpose of the study is to investigate methodological aspects of a new method that can be useful for the investigation of periglacial environments, and demonstrate its feasibility. Therefore, we make an important step towards quantitative usage, such as the estimation of ice content.
The main purpose is not, however, to actually calculate ice content and compare the results with ground truth data. We admit that this would be desirable to have, but it is difficult to obtain in general, and not available for our test sites. We believe that our results are nevertheless important and interesting for a broad readership.

C: We will restructure the abstract and introduction to better describe the purpose of the study. We will also provide a better context of existing methods and research and explain the potential improvement by our method (see (1)(2)(3)). However, instead of a full literature review, we will prefer to refer to a small selection, as the importance of ground ice, and the usefulness of geophysical methods in general, and electrical methods in particular, have already been discussed in textbooks.
There is not much additional information about the field sites (see (6)(7)), but we will try to give a better explanation of what we know and how we can compare the data with existing information.

---

## Author Comment (AC2) · 14 Jun 2019

**Review 2 (02.05.2019)**

**R**: Referee's comment

**A**: Author's response

**C**: Change in manuscript

**R:** SUMMARY
This paper presents data and modelling results for broadband spectral capacitive resistivity experiments performed in cold regions. The experiments appear sound in design, and the data are novel and very interesting from the perspective of electrical/electromagnetic geophysics and cold-regions research.

**A**: We appreciate the overall positive evaluation and we thank for the constructive comments.

**(1)**
**R:** However, the paper has a few shortcomings. The objectives of the paper are not entirely clear at first. Is the focus on SIP or CR? It becomes clear (I think) that the focus is on cold-regions application of broadband spectral CR. If this is the desired focus, the paper would be made more impactful by including: 1) a review of electrical, IP, SIP, and CR applications to cold regions and permafrost; 2) a clear description of the benefits, limitations, and favourable conditions for broadband spectral CCR, and how these relate to cold regions; and 3) a more thorough analysis of the inversion results in terms of cold-regions ground properties of interest such as water content, ice content and temperature.

**A:** The CCR (in the spectral way we use it) is actually the method of SIP in a higher frequency range, additionally using the capacitive coupling instead of the "normal" galvanical coupling. Without the capacitive coupling, the method would be similar to Grimm & Stillman (2015) who call it "Broadband SIP" or the "HFIP" (high-frequency IP) from Zorin & Ageev (2017). We focus on CCR as we used it in the field with the logistical advantages and in the terms of the discussed electrode height effect.
**C:** 1) A review of existing methods and studies of ERT, IP, CCR will be provided in the introduction. We will, however, refer back to textbooks instead of a full literature review, because some of the aspects have already been summarized.
2) Our CCR method will be described more precisely.
3) The analysis of the Inversion results will be described and compared with external information in more detail (p13 ff.). We will not be able to provide an extensive ground-truth comparison, however, because ground truth, for example in terms of ice content, is not available for our test sites. Since the focus of the paper is more on methodological aspects, we hope that this issue is not so critical.

**(2)**
**R:** Abstract: Lacks focus on objective and results.

**A:** We agree that the abstract needs to be rewritten.
**C:** Abstract will be reformulated to describe objective and results more clearly.

**(3)**
**R:** p1,L19: Why? High conductivity material exhibit spectral characteristics as well.

**A:** It is true that high cond. subsurface exhibits spectral characteristics. But for the determination of the permittivity (in addition to resistivity), lower cond. material is more suitable, because we are limited in frequency range. The reason is, that as lower the conductivity is, the transition to displacement currents and therefore the possible extraction of the permittivity, happens for lower frequencies (e.g. Zorin & Ageev (2017)).
**C:** The relationship between conductivity and determination of permittivity will become clear in a new section explaining the operating range off CCR (see below).

**(4)**
**R:** p2,L28: What about Routh et al. (1998), Kemna et al. (2000) and several works thereafter?

**A:** These inversions of IP data works in the way that they invert single frequency data and the in postprocessing integrate all spectral and spatial data, which is the difference to the work of Günther and Martin (2016) and Maurya et al. (2018), who invert all frequencies at the same time. But we agree, that this works should be mentioned in the manuscript.
**C:** The works with an explanation will be included in the introduction.

**(5)**
**R:** p.2,L30: Introduction is somewhat unclear. It starts out focussed on SIP, then CCR, but then states the objective as investigating the field applicability of [spectral] CCR in cold regions. To support the latter, the intro needs a little more background on electrical geophysics and material properties in permafrost and/or glaciology.

**A:** We agree.
**C:** The introduction will be reorganized. There will be a review of existing electrical methods and studies before explaining what is new and what is the goal of our method (see (1)). Additionally, the connection to material properties of ice will be made clearer.

**(6)**
**R:** eq.1: Although somewhat semantic, I view the low-frequency CCR experiment as responding to the complex conductivity where the imaginary conductivity has a contribution from the real dielectric. Of course, at higher frequency and in the presence of ice, permittivity may be more relevant. However, I do not think that equation 1 should be referred to as representing the "complex permittivity." Consider "effective permittivity" which has a contribution from the imaginary conductivity. The distinction is important because the true dielectric permittivity is what results in wave propagation in Maxwell's equations, not the imaginary conductivity. Furthermore, to talk only of displacement currents denies the possibility of IP-type currents which may dominate at lower frequency.

**A:** We agree, and actually tried to express this by using the term "effective value" on p5., l.11.
**C:** We will now explicitly call eq. (1) "effective permittivity", and will also add a few lines on conduction and displacement currents, refering to the overview given by Loewer et al. (2017).

**(7)**
**R:** p.6,L4: You say the "conduction current" is in-phase, but then you say that IP is concerned with the conduction current part of the impedance. You need to be clearer on the distinction between the real conductivity (conduction current), the imaginary

conductivity (IP current), and the real permittivity (displacement current).
eq.4: Again, consider noting that any IP effect will be wrapped up in here as εeff = εR + σI/ω.

**A:** see (6)
**C:** We will introduce that in a short paragraph (see (6)).

**(8)**
**R:** p.7,L8: The height effect is also discussed by Wang et al. (2016) but in a shady pseudojournal. The authors could decide if it warrants consideration.

**A:** Thank you for pointing this out to us. We agree that it is relevant for our work.
**C:** The reference will be included.

**(9)**
**R:** p.8,L3: You say the inversion is frequency dependent, but then go on to say that the system response is controlled by geometry, not frequency. Clarify.

**A:** The statement refers to the fact that we work in the range of "geometric sounding" (same as ERT) and so we do not have a frequency sounding (as e.g. RMT); See (10)
**C:** see (10)

**(10)**
**R:** p.8,L5: You need to thoroughly describe the "operating range" of CCR with respect to treatment of the data for the single site inversion and the 2D inversion. Application of a 2D resistivity inversion (with geometry-based sensitivity) requires low-induction number conditions (which actually appears to be violated for some of your lower frequency-resistivity combinations). Does the single-site inversion require LIN conditions? What about wave effects? For some of the high frequency-resistivity pairs encountered, quasistatic conditions are violated and a true permittivity will result in wave propagation. This should not(?) affect CC model fits, but it should(?) affect the 2D inversions using a resistivity-type sensitivity function.

**A:** The method of CCR operates in the physical range of "geometric sounding" (GS), which indeed requires LIN conditions to be fulfilled. We do not agree, however, that these are violated for some of our data (see more detailed explanation below). Quasistatic conditions, however, are not necessary, because it is the wavenumber that determines which physical process dominates. The detailed explanation is as follows:

The physical boundaries are given by the effects of electromagnetic induction (EMI) and wave propagation (WP). This consideration of the three processes is described in Weidelt (1997) and the "parameter range" refers to the frequency f, the spatial scale a (i.e. distance between transmitter and receiver), electrical conductivity and permittivity.

The equation that allows to compare the processes is: $\gamma^2 = \frac{4\pi}{a^2} + \frac{2i}{\delta^2} + \frac{4\pi}{\lambda^2} = GS + EMI + WP$, where the process corresponding to the largest term will dominate, and the other terms may be negligible, depending on their magnitude.

Skin depth: $\delta = \dfrac{1}{\sqrt{\pi f \mu \frac{1}{\rho}}}$ ; wave length $\lambda = \dfrac{2\pi}{\omega \sqrt{\varepsilon\mu}}$

If we calculate this as a worst-case for our maximum frequency 240 kHz, a smallest resistivity (lowest conductivity) of 100 Ohmm and a spatial length of 10 m, we are still clearly in the range of GS. (GS=4; EMI = 0.02; WP=8e-5)

So we think that there is no violation neither to induction effects nor to wave propagation. For the presented measurements, we always have low-induction numbers and therefore can neglect possible resulting problems in terms of the inversion.

Anyway, we agree that this point has to made clearer in the manuscript.

**C:** We will describe the operating range and the topic of GS, EMI and WP in a new section within chapter 3.

**(11)**
**R:** p8,L6: "Induction effects" would typically be understood to mean inductive source effects of current-carrying cables. You don't have these. So, do you mean magnetic coupling as described by McNeill?

**A:** Yes, we (as in Fiandaca (2018)) mean the electromagnetic induction effects.
**C:** Will be described more deeply (see (10), and we will use more precise terminology).

**(12)**
**R:** p.13,L25: Well, the dielectric constants for rock and snow are both around 3-5, so...

**A:** Yes, this is the reason why the values are so close and material is difficult to distinguish.
**C:** We will include this in the statement (p13 L25).

**(13)**
**R:** p.13,L32: Snow cover typically inhibits frozen ground.

**A:** We agree, the sentence is unclear and actually not necessary for the further discussion.
**C:** Will change the statement.

**(14)**
**R:** Table 1: Add water. More discussion is required in comparing recovered values to expected material properties.

**A:** The discussion can be done in more detail and by including water in table 1 and the discussion.
**C:** Will add water in table 1 and deepen the discussion when comparing literature values with the inversion results for the test sites (p15 L4 ff. and p17 L5 ff.).

**(15)**
**R:** p.15,L5: In comparing to literature values, what about the observations by Weigand and Kemna (2016) that SIP model parameters obtained from a CC model are biased? Is this alleviated by having c as a free parameter and/or by having 19 points in the spectrum?

**A:** We thank for pointing this out and agree that this should be included in the text. Yes, we think that the fact of having c as free parameter (see below (18)) and having the high density of 19 frequency measurements in the range of estimated tau alleviate the effect of bad determined CC parameters. The inverted tau-value are in the expected range of relaxation for ice and snow, what strengthen our assumption.
**C:** Observations of Weigand and Kemna (2016) will be included and discussed in the context of our results.

**(16)**
**R:** Fig.8: Use same scales as Fig.7. Are some of the observed differences attributable to height effects or breakdown of LIN conditions?

**A:** As explained there should be no breakdown of LIN conditions (see (10)). Moreover the height effects in our cases should be too small to influence the 2D inversion, as shown in fig. 6 and discussion.  The differences have to be due to the fact that figure 8 is a pseudosection, whereas figure 7 is a 2-D inversion result. We actually had devoted an extra paragraph to this discussion (p. 13, l. 10ff).

**C:** Since spatial variation in pseudosections is always smoother compared to inversion results, using the same scales would result in loss of information. Therefore, we would prefer to leave the scales as they are. Instead, we try to clarify the discussion of pseudosection and inversion result.

**(17)**
**R:** p.18,L5: Why is the DC permittivity so (unreasonably) high?

**A:** The values are higher than expected comparing with literature values. Nevertheless, LF values of permittivity for natural water, specially with high salinity (as it is in the lake), can be very high (Seshadri et al, 2008). On the other hand an overestimation may be caused by the measured low frequency values (see e.g. Zorin and Ageev (2017)).
**C:** The role of epsDC and the high inversion results will be discussed in more detail.

**(18)**
**R:** p.18,L11: Is there any benefit to setting c constant (i.e., choosing a decomposition for the CC model). Is it reasonable for c to show so much variability? Is it highly sensitive, and if so, is it just absorbing error in the inversion?

**A:** We did some inversion tests with constant c. It shows that a constant c cannot fit the data of all the different spectra. We think that the different subsurface materials and conditions explain the variability of c and justify the use as free parameter.
**C:** Will be mentioned in the text.

**(19)**
**R:** p.18,L19: Actually, the LF permittivity of water is around 80, but you need to get up to 1010 to 1012 Hz before it drops to around 3.

**A:** We have to make this clear that the relaxation of water refers to higher frequencies and the value of 80 is the low-frequency value.
**C:** Will be changed in the text and described in context of adding literature values of water in table 1 (see (14)).

**(20)**
**R:** Figure 10: a) Use dash and solid. b) What is the distinction between black and purple lines?

**A:** b) The distinction is measured (purple) and inverted (black)
**C:** a) will be changed b) will be added in the figure caption

New References:

M. Loewer, T. Günther, J. Igel, S. Kruschwitz, T. Martin and N. Wagner: Ultra-broad-band electrical spectroscopy of soils and sediments – a combined permittivity and conductivity model; Geophysical Journal International 210, 1360-1373; 2017